# PID Control-Based Self-Healing to Improve the Robustness of Large Language Models

**Zhuotong Chen**                                                                    *ztchen@ucsb.edu*
*Department of Electrical and Computer Engineering, University of California, Santa Barbara*

**Zihu Wang**                                                                        *zihu_wang@ucsb.edu*
*Department of Electrical and Computer Engineering, University of California, Santa Barbara*

**Yifan Yang**                                                                       *yifanyang@ucsb.edu*
*Department of Computer Sciences, University of California, Santa Barbara*

**Qianxiao Li**                                                                      *qianxiao@nus.edu.sg*
*Department of Mathematics, National University of Singapore, Singapore*
*Institute of High Performance Computing, A\*STAR, Singapore*

**Zheng Zhang**                                                                      *zhengzhang@ece.ucsb.edu*
*Department of Electrical and Computer Engineering, University of California, Santa Barbara*

**Reviewed on OpenReview:** *https://openreview.net/forum?id=Fu4mwBOXIU*

## Abstract

Despite the effectiveness of deep neural networks in numerous natural language processing applications, recent findings have exposed the vulnerability of these language models when minor perturbations are introduced. While appearing semantically indistinguishable to humans, these perturbations can significantly reduce the performance of well-trained language models, raising concerns about the reliability of deploying them in safe-critical situations. In this work, we construct a computationally efficient self-healing process to correct undesired model behavior during online inference when perturbations are applied to input data. This is formulated as a trajectory optimization problem in which the internal states of the neural network layers are automatically corrected using a PID (Proportional-Integral-Derivative) control mechanism. The P controller targets immediate state adjustments, while the I and D controllers consider past states and future dynamical trends, respectively. We leverage the geometrical properties of the training data to design effective linear PID controllers. This approach reduces the computational cost to that of using just the P controller, instead of the full PID control. Further, we introduce an analytical method for approximating the optimal control solutions, enhancing the real-time inference capabilities of this controlled system. Moreover, we conduct a theoretical error analysis of the analytic solution in a simplified setting. The proposed PID control-based self-healing is a low-cost framework that improves the robustness of pre-trained large language models, whether standard or robustly trained, against a wide range of perturbations. A detailed implementation can be found in:`https://github.com/zhuotongchen/PID-Control-Based-Self-Healing-to-Improve-the-Robustness-of-Large-Language-Models`.

## 1 Introduction

The growth of data and advancements in computing power have heralded a new era in the field of natural language processing (NLP), significantly shaped by the advent of deep neural networks. One of the most influential innovations in this domain is the transformer architecture (Vaswani et al., 2017), which is the

fundamental block of many successful large language models (LLMs) (Brown et al., 2020). This architecture has become the state-of-the-art in various NLP tasks, including sentiment analysis (Vinodhini & Chandrasekaran, 2012), text summarization (Nenkova & McKeown, 2012), and speech recognition (Hannun et al., 2014), among others.

However, many deep neural networks are vulnerable to malicious perturbations (Morris et al., 2020). While appearing semantically indistinguishable to humans, these perturbations can significantly degrade the performance of pre-trained LLMs. This vulnerability raises concerns about the reliability of deploying them in safety-critical situations, such as in clinical decision support systems, where LLMs serve a critical role in assisting healthcare professionals with patient care insights (Huang et al., 2019). In response to this challenge, there has been significant progress in developing algorithms to enhance model robustness against such perturbations (Yoo & Qi, 2021; Zhu et al., 2019; Wicker et al., 2021). Predominantly, these methods are rooted in the foundation of adversarial training (Madry et al., 2018), a method where pre-trained LLMs are fine-tuned (or trained from random initialization) to overcome the effects of specific adversarial perturbations. This is achieved by adjusting either the entire set of model parameters or a significant portion thereof (Hu et al., 2021). Despite its effectiveness, this approach raises three critical concerns. Firstly, adjusting model parameters using adversarial examples requires substantial computational resources (Zhang et al., 2019). Due to the discrete input space of NLP tasks, searching for an adversarial example generally involves solving a combinatorial optimization problem (Bernhard & Vygen, 2008), which suffers from an exponential growth in the number of feasible solutions as the size of the problem increases. Secondly, there exists a potential trade-off where improved adversarial robustness may lead to compromised performance on standard, natural datasets (He et al., 2021). Thirdly, and more problematically, adversarial training is less effective against unforeseen adversarial perturbations (Tramer & Boneh, 2019). This limitation becomes particularly noticeable when deploying LLMs in practice, where anticipating the potential adversarial attacks in advance is nearly impossible.

In this paper, we investigate the concept of a self-healing process as a cost-effective method to improve the robustness of pre-trained LLMs against a range of perturbations. The most well-known self-healing mechanism is probably the human immune system: B cells and T cells can work together to identify and kill many external attackers (e.g., bacteria) to maintain the health of the human body (Rajapakse & Groudine, 2011). In the context of machine learning, self-healing refers to the ability of a model to automatically identify and correct issues that may arise during its operation (Wang et al., 2021; Chen et al., 2022). To achieve this, we consider a pre-trained LLM (typically a composition of transformation blocks) as a discretization of the continuous dynamical system (E, 2017), this allows us to formulate the robustness issue of LLMs as a trajectory optimization problem (Hehn & D'Andrea, 2015). Our approach involves designing PID (Proportional-Integral-Derivative) controllers at hidden layers of a pre-trained LLM. A PID controller continuously calculates an error value as the difference between a desired reference and a measured process variable and applies a correction control signal based on proportional, integral, and derivative terms. More specifically, let the error value be the difference between a desired reference and the current state. If the error is large, the output of the P controller will be proportionately large, thereby making a significant adjustment and helping the controller respond quickly to errors. The I controller determines the present control output based on the integration of past errors, which ensures that even small errors are corrected over time. The D controller generates control signals based on the derivative of the error dynamics. The combination of P, I, and D controllers quantifies undesired model behavior from present errors, past accumulated errors, and future error trends, and generates control signals to correct the errors. Figure 1 illustrates the proposed PID control-based self-healing framework. Given a $T$-layer LLM, time-dependent PID controllers (represented as $P_t$, $I_t$, and $D_t$) generate a feedback control based on the state $\mathbf{x}_t$ (to simplify the demonstration, only $\mathbf{x}_t$ is considered as the input for both I and D controllers). These feedback controls aim to remove the undesirable effects caused by input perturbations.

The methodology of constructing a self-healing process to improve the robustness of deep neural networks was initially introduced in Chen et al. (2020) and its subsequent work Chen et al. (2022). It leveraged a closed-loop control method to detect and correct potential errors applied to input data. This method belongs to a special case of P control in the proposed PID control framework, in which only the errors from the present states are corrected. However, the effectiveness of using only proportional controllers is

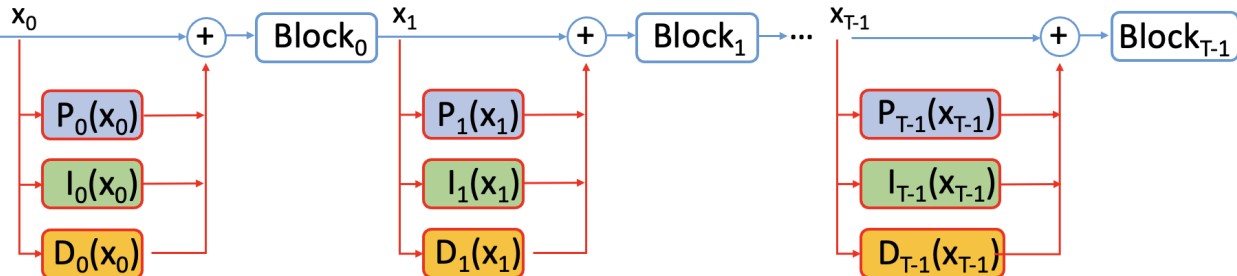

Figure 1: The structures of feed-forward deep neural network (highlighted in blue) and the proposed PID control method (highlighted in red).

limited, as it merely addresses the error at each time step, neglecting the overall error dynamics (we provide numerical evidence to show this in Section 3.4). Additionally, a major limitation of the closed-loop control method adapted in Chen et al. (2020; 2022) is its computational inefficiency. The optimal control solution requires simulating the Hamiltonian dynamics over several iterations during online inference (Pontryagin, 1987), which involves both forward and backward propagation of a deep neural network (Chen et al., 2020) (we discuss the details about simulating the Hamiltonian dynamics in Section 2.1). This inefficiency renders the self-healing framework impractical for deployment in large-scale LLMs, which may contain millions or even billions of parameters (Kenton & Toutanova, 2019; Liu et al., 2019; Brown et al., 2020). To address the challenges associated with the previously mentioned adversarial training and existing closed-control method, this study presents three contributions:

- We introduce a novel PID control framework to realize the self-healing capability to improve the robustness of LLMs during online inference. The proposed framework generalizes the conventional robustness improvement methods that predominantly focus on proportional errors. We demonstrate that employing all P, I, and D controllers can be as computationally efficient as single control schemes, achieved through special controller design.

- We approximate the layer-wise transformations in the pre-trained LLM as linear orthogonal transformations and derive an analytical solution for generating PID control solutions. This analytical method yields a closed-form expression for the optimal solution, enhancing the speed of online inference. This acceleration is especially beneficial when integrating the self-healing framework into LLMs. While these approximations might not align with practical scenarios, our method exhibits superior robustness improvement in a variety of numerical experiments.

- We derive a comprehensive error analysis of the controlled system, highlighting the robustness improvement of LLMs through PID control solutions. This analysis provides insight into the effectiveness of PID control in improving the robustness of LLMs in simplified settings, thereby contributing to the understanding of controlled systems.

## 1.1 Background on PID Control

Here we provide background knowledge on PID control, which is an essential building block of our proposed framework. A PID (Proportional, Integral, Derivative) controller is a feedback-based control system extensively utilized in industrial settings and numerous other domains where continuous adjustment is essential. Specifically, given a continuous dynamic system, typically described as an ordinary differential equation,

$$\dot{\mathbf{x}}_t = \Psi(\mathbf{x}_t, \mathbf{u}_t), \quad \mathbf{x}_0 \in \mathbb{R}^d,$$

where $\dot{\mathbf{x}}_t$ represents the derivative with respect to time, while $\Psi : \mathbb{R}^d \times \mathbb{R}^m \to \mathbb{R}^d$ defines a function or vector field. The term $\mathbf{x}_0$ specifies the initial state. Moreover, $\mathbf{u}_t \in \mathbb{R}^m$ denotes the control input exerted on the dynamic system (e.g. in this work, the control input is applied linearly to the state $\mathbf{x}_t$). The PID control

continuously computes an error term, $e_t$, by calculating the difference between a target reference $\mathbf{r}_t$ and the actual state variable $\mathbf{x}_t$,

$$e_t = \|\mathbf{r}_t - \mathbf{x}_t\|.$$

The construction for the target reference generally depends on the application. In this work, the target reference is constructed by a sequence of embedding manifolds (see Section 2.1), and the process variable represents the hidden state of a deep neural network during forward propagation.

Then, a PID controller applies a control $\mathbf{u}_t$ based on proportional, integral, and derivative terms to correct the measured error. In the continuous case,

$$\mathbf{u}_t = K_p e_t + K_i \int_0^t e(\tau)d\tau + K_d \frac{de_t}{dt},$$

where $K_p$, $K_i$, and $K_d$, all non-negative, denote the coefficients for the proportional, integral, and derivative terms respectively. In the PID control design,

- The output from proportional control directly correlates with the current value of the error, $e_t$. Thus, a larger error will yield a proportionally larger control output, adjusted by the gain factor $K_p$. However, employing only proportional control leads to a persistent difference between the desired target and the actual process variable, since the generation of the proportional response necessitates the presence of an error.

- The output from integral control considers the accumulation of past error values over time. This means that when a residual error remains after proportional control is applied, the integral control works to correct this residual error by leveraging the historical total error. This will result in the proportional effect diminishing as the error decreases, but this is compensated for by the growing integral effect.

- The output from derivative control provides an estimate of the trend of the error, using its current rate of change as a basis. It effectively seeks to reduce the effect of the error by exerting a control influence generated by the rate of error change. Hence, the more rapid the error's progression, the more intense the applied correction.

Although a PID controller has three control terms, some applications need only one or two terms to provide appropriate control. This is achieved by setting the unused parameters to zero and is called a PI, PD, P, or I controller in the absence of the other control actions. PD controllers are fairly common in applications where integral action would be sensitive to measurement noise, but the derivative term is often needed for the system to reach its target reference.

## 2    The PID Control-Based Self-Healing Framework for Large Language Models

In this work, we use the concept of "self-healing" to describe the capability of an LLM to automatically correct errors that may arise. This idea has been studied in the domain of integrated circuits, primarily addressing errors attributable to variations in nano-scale fabrication processes. More specifically, the internal dynamics of an electronic circuit network can be understood through the lens of ordinary differential equations (Ho et al., 1975). This approach conceptualizes the circuit network in terms of state variables that track nodal voltages and branch currents as they change over time. Moreover, it's possible to create a self-healing system within the circuit network, enabling it to continuously monitor and optimize its performance throughout the lifetime of operation (Tang et al., 2012; Lee et al., 2012). There is a growing body of research, including works by (E, 2017; Haber & Ruthotto, 2017; Li et al., 2018b), and others, demonstrating the connection between dynamical systems and deep neural networks. In our study, we interpret a pre-trained $T$-layer LLM as a form of discrete dynamical system,

$$\mathbf{x}_{t+1} = F_t(\mathbf{x}_t + \pi_t(\mathbf{x}_t), \boldsymbol{\theta}_t), \ \ \forall t = 0, 1, ..., T-1, \tag{1}$$

where $F_t(\cdot, \boldsymbol{\theta}_t) : \mathbb{R}^d \to \mathbb{R}^d$ represents a transformer block parametrized by $\boldsymbol{\theta}_t$, $\pi_t : \mathbb{R}^d \to \mathbb{R}^d$ is a feedback controller that maps the current state $\mathbf{x}_t$ to a control action. We aim to construct feedback controllers $\overline{\pi} := \{\pi_t\}_{t=0}^{T-1}$ to ensure that the controlled states $(\mathbf{x}_t + \pi_t(\mathbf{x}_t))$ yield the desired output when perturbations are applied to input data. This can be formulated as a trajectory optimization problem,

$$\min_{\overline{\pi}} \mathbb{E}_{(\mathbf{x}_0, y) \sim \mathcal{D}} \left[ J(\mathbf{x}_0, \mathbf{y}, \overline{\pi}) \right] := \min_{\overline{\pi}} \mathbb{E}_{(\mathbf{x}_0, y) \sim \mathcal{D}} \left[ \Phi(\mathbf{x}_T, y) + \sum_{t=0}^{T-1} \mathcal{L}(\{\mathbf{x}_s\}_{s=0}^t, \pi_t, f_t) \right], \text{ s.t. } equation\ 1 \qquad (2)$$

where initial states and labels $(\mathbf{x}_0, y)$ are sampled from the underlying data distribution $\mathcal{D}$. The terminal loss $\Phi(\mathbf{x}_T, y)$ evaluates the discrepancy between the terminal state and a pre-defined destination set. In machine learning applications, this measures the consistency between the terminal state $\mathbf{x}_T$ (or its transformation) with the true label (e.g., cross-entropy loss). However, this is not feasible in general machine learning applications as the true label $y$ remains unknown during inference. More specifically, during online inference, given an initial condition $\mathbf{x}_0$, its label $y$ cannot be accessed to optimize the states since this quantity is unknown. Consequently, the terminal loss is negated by setting it to zero. In these cases, the optimal controllers $\{\pi_t\}_{t=0}^{T-1}$ minimize the cumulative running losses $\mathcal{L}(\{\mathbf{x}_s\}_{s=0}^t, \pi_t, f_t)$, which assess the state trajectory and control using certain measurement function $f_t$.

In Section 2.1, we construct the running loss, which forms a crucial part of the objective function defined in equation 2. This objective function needs to solve a complex optimization problem for each initial state during online inference, which presents a challenge to computational efficiency. To address this, the subsequent Section 2.2 presents a more efficient algorithm for solving the objective function under specific assumptions. Furthermore, Section 2.3 presents a comprehensive theoretical error analysis for the proposed algorithm. Lastly, Section 2.4 provides additional details on the implementation of constructing PID controls.

## 2.1 PID Control Design via Embedding Manifolds

In analyzing an LLM through the lens of discrete dynamical systems, we observe that its state trajectory, governed by the composition of transformations, forms a lower-dimensional structure embedded in the ambient state space, also known as the "manifold hypothesis" (Fefferman et al., 2016) (empirical evidence is shown in Table 7). This can be conceptualized as a sequence of embedding manifolds. We consider a r-dimensional smooth manifold embedded in $\mathbb{R}^d$ as $\{\mathbf{x} : f(\mathbf{x}) = \mathbf{0}\}$, where $f : \mathbb{R}^d \to \mathbb{R}^{(d-r)}$ is a surjective mapping that can be used to measure the distance between a state $\mathbf{x}_t$ to the embedding manifold. For instance, $\|f(\mathbf{x})\| = 0$ if $\mathbf{x}$ belongs to the embedding manifold, and $\|f(\mathbf{x})\| > 0$ if $\mathbf{x}$ is outside the embedding manifold.

At each time step $t$ (e.g., the $t^{\text{th}}$ hidden states), we construct three embedding manifolds with three distinct surjective functions $f_t^P : \mathbb{R}^d \to \mathbb{R}^{(d-r)}$, $f_t^I : \mathbb{R}^d \to \mathbb{R}^{(d-r)}$, and $f_t^D : \mathbb{R}^d \to \mathbb{R}^{(d-r)}$, that represent the embedding manifolds of the state, the integration of past states, and the derivative of the state, respectively. In a discrete setting, integration corresponds to the accumulation of past states, while the derivative is approximated by the difference between two successive states. Under this setting, $f_t^I$ denotes the embedded manifold of past states, and $f_t^D$ represents the embedded manifold derived from the difference between two consecutive states. Given these embedding functions, we propose the following running loss to evaluate the controlled state at time step $t$,

$$\mathcal{L}(\{\mathbf{x}_s\}_{s=0}^t, \pi_t, (f_t^P, f_t^I, f_t^D)) := \frac{1}{2} \|f_t^P(\mathbf{x}_t + \pi_t(\mathbf{x}_t))\|_2^2 + \frac{1}{2} \|f_t^I(\mathbf{x}_t + \pi_t(\mathbf{x}_t) + \sum_{s=0}^{t-1} \mathbf{x}_s)\|_2^2$$

$$+ \frac{1}{2} \|f_t^D(\mathbf{x}_t + \pi_t(\mathbf{x}_t) - \mathbf{x}_{t-1})\|_2^2 + \frac{c_t}{2} \|\pi_t(\mathbf{x}_t)\|_2^2, \qquad (3)$$

where the layer-dependent regularization term $c_t$ prevents using large controls. The running loss consists of three components, each assessing the error in the controlled state through distinct embedding functions: proportional, integration, and derivative. In this construction of running loss, the optimal controller results in a controlled state $(\mathbf{x}_t + \pi_t(\mathbf{x}_t))$ which is expected to closely align with the state embedding manifold, evaluated by $f_t^P$. Additionally, the controller must ensure that past controlled states remain close to the embedding manifold of integrated states, as evaluated by $f_t^I$, and the state's derivative should similarly align closely with the manifold of the state's derivative embedding, as quantified by $f_t^D$.

The objective function defined in equation 2 and the associated running loss detailed in equation 3 can be solved via the dynamical programming principle (Bellman, 1952). However, this method faces exponential complexity in terms of the dimension of the state. To overcome this "curse of dimensionality", we can reinterpret the optimal control problem through Pontryagin's Maximum Principle and approximate it using the method of successive approximation (Chen et al., 2020). To begin with, we define the Hamiltonian $H(t, \{\mathbf{x}_s\}_{s=0}^t, \mathbf{p}_{t+1}, \boldsymbol{\theta}_t, \mathbf{u}_t)$ as

$$H(t, \{\mathbf{x}_s\}_{s=0}^t, \mathbf{p}_{t+1}, \boldsymbol{\theta}_t, \mathbf{u}_t) := \mathbf{p}_{t+1}^T \cdot F_t(\mathbf{x}_t + \mathbf{u}_t, \boldsymbol{\theta}_t) - \mathcal{L}(\{\mathbf{x}_s\}_{s=0}^t, \mathbf{u}_t, (f_t^P, f_t^I, f_t^D)),$$

where $\mathbf{u}_t = \pi_t(\mathbf{x}_t)$ is a control solution. Pontryagin's maximum principle consists of a two-point boundary value problem,

$$\mathbf{x}_{t+1}^* = \nabla_p H(t, \{\mathbf{x}_s\}_{s=0}^t, \mathbf{p}_{t+1}, \boldsymbol{\theta}_t, \mathbf{u}_t), \qquad (\mathbf{x}_0, y) \sim \mathcal{D}, \qquad (4)$$

$$\mathbf{p}_t^* = \nabla_x H(t, \{\mathbf{x}_s\}_{s=0}^t, \mathbf{p}_{t+1}, \boldsymbol{\theta}_t, \mathbf{u}_t), \qquad \mathbf{p}_T = \mathbf{0}, \qquad (5)$$

plus a maximization condition of the Hamiltonian.

$$H(t, \{\mathbf{x}_s\}_{s=0}^t, \mathbf{p}_{t+1}, \boldsymbol{\theta}_t, \mathbf{u}_t^*) \geq H(t, \{\mathbf{x}_s\}_{s=0}^t, \mathbf{p}_{t+1}, \boldsymbol{\theta}_t, \mathbf{u}_t), \ \forall \ \mathbf{u}_t \ \text{and} \ \forall t \in \mathcal{T}. \qquad (6)$$

To obtain a numerical solution, one can consider iterating through the forward dynamic equation 4 to obtain all states $\{\mathbf{x}_t\}_{t=0}^{T-1}$, the backward dynamic equation 5 to compute the adjoint states $\{\mathbf{p}_t\}_{t=0}^{T-1}$, and maximizing the Hamiltonian defined in equation 6 with current states and adjoint states via gradient ascent. This iterative process is continued until convergence. Given an initial condition $\mathbf{x}_0$, Pontryagin's Maximum Principle characterizes the optimal feedback control $\pi_t(\mathbf{x}_t)$ with an open-loop control $\mathbf{u}_t$, this open-loop control necessitates both forward and backward propagation through a pre-trained deep neural network over several iterations during online inference.

However, implementing the above Pontryagin's Maximum Principle is generally infeasible for LLMs. In the subsequent section, we construct an analytic solution with certain relaxation assumptions.

## 2.2 An Analytic Solution for Fast Inference

In this section, we develop an analytic solution for solving the objective function defined in equation 2, under certain assumptions. These assumptions are summarized in the following,

- **Assumption 1:** Both embedding manifold and layer-wise transformation are simplified as linear functions. In this case, the layer-wise transformation, denoted as $F_t(\cdot)$, is linearized through a matrix $\boldsymbol{\theta}_t \in \mathbb{R}^{d \times d}$. A smooth embedding manifold is represented by a linear embedding subspace. This linear embedding subspace is defined by a set of basis vectors, which are captured by the column space of a matrix $\mathbf{V} \in \mathbb{R}^{d \times r}$, corresponding to an $r$-dimensional embedding subspace.

- **Assumption 2:** Both embedding manifold and layer-wise transformation are orthogonal. In this case, layer-wise transformations are represented by orthogonal matrices, satisfying $\boldsymbol{\theta}_t^\top \boldsymbol{\theta}_t = \boldsymbol{\theta}_t \boldsymbol{\theta}_t^\top = \mathbf{I}$. Additionally, the basis vectors $\mathbf{V}_t^P$, $\mathbf{V}_t^I$, and $\mathbf{V}_t^D$ are considered to be mutually orthogonal,

$$(\mathbf{V}_t^P)^\top \mathbf{V}_t^I = \mathbf{0}, \quad (\mathbf{V}_t^P)^\top \mathbf{V}_t^D = \mathbf{0}, \quad (\mathbf{V}_t^I)^\top \mathbf{V}_t^D = \mathbf{0}.$$

Based on these assumptions, the computational costs of the proposed control algorithm are similar to performing forward propagation with the original model. This implies that the PID control approach introduces negligible computational cost. The negative impact of these assumptions are discussed in Section 3.4.

**An analytic solution under Assumption 1.** In the special linear case, for the linear embedding subspaces linked to the state, state integration, and state derivative, we define the basis as $\mathbf{V}_t^P$, $\mathbf{V}_t^I$, and $\mathbf{V}_t^D$, respectively. Consequently, the embedding manifolds, represented by the surjective functions $f_t^P$, $f_t^I$, and $f_t^D$, are orthogonal projections $\mathbf{Q}_t^P$, $\mathbf{Q}_t^I$, and $\mathbf{Q}_t^D$, where

$$\mathbf{Q}_t^P = \mathbf{I} - \mathbf{V}_t^P(\mathbf{V}_t^P)^\top, \quad \mathbf{Q}_t^I = \mathbf{I} - \mathbf{V}_t^I(\mathbf{V}_t^I)^\top, \quad \mathbf{Q}_t^D = \mathbf{I} - \mathbf{V}_t^D(\mathbf{V}_t^D)^\top.$$

The following proposition solves the objective function defined in equation 2 under linearity assumptions.

**Proposition 1.** *Consider the following objective function,*

$$\min_{\overline{\pi}} \mathbb{E}_{(\mathbf{x}_0, y) \sim \mathcal{D}} \left[ J(\mathbf{x}_0, y, \overline{\pi}) \right] := \min_{\overline{\pi}} \mathbb{E}_{(\mathbf{x}_0, y) \sim \mathcal{D}} \left[ \Phi(\mathbf{x}_T, y) + \sum_{t=0}^{T-1} \mathcal{L}(\{\mathbf{x}_s\}_{s=0}^t, \pi_t, (\mathbf{Q}_t^P, \mathbf{Q}_t^I, \mathbf{Q}_t^D)) \right],$$

$$\text{s.t. } \mathbf{x}_{t+1} = \boldsymbol{\theta}_t(\mathbf{x}_t + \pi_t(\mathbf{x}_t)). \tag{7}$$

*the optimal value function, parametrized as $V(\mathbf{x}_t) = \mathbf{x}_t^\top \mathbf{P}_t \mathbf{x}_t$, satisfies the Riccati equation:*

$$\mathbf{P}_t = \frac{1}{2}\mathbf{Q}_t + \boldsymbol{\theta}_t^\top \mathbf{P}_{t+1}\boldsymbol{\theta}_t - \frac{1}{2}(\mathbf{Q}_t + 2\boldsymbol{\theta}_t^\top \mathbf{P}_{t+1}\boldsymbol{\theta}_t)^\top (\mathbf{Q}_t + 2\boldsymbol{\theta}_t^\top \mathbf{P}_{t+1}\boldsymbol{\theta}_t + c_t\mathbf{I})^{-1}(\mathbf{Q}_t + 2\boldsymbol{\theta}_t^\top \mathbf{P}_{t+1}\boldsymbol{\theta}_t). \tag{8}$$

*The optimal control solution is given by*

$$\pi_t(\mathbf{x}_t) = -(\mathbf{Q}_t + c \cdot \mathbf{I} + 2\boldsymbol{\theta}_t^\top \mathbf{P}_{t+1}\boldsymbol{\theta}_t)^{-1}(\mathbf{Q}_t + 2\boldsymbol{\theta}_t^\top \mathbf{P}_{t+1}\boldsymbol{\theta}_t)\mathbf{x}_t, \tag{9}$$

*where $\mathbf{Q}_t = \mathbf{Q}_t^P + \mathbf{Q}_t^I + \mathbf{Q}_t^D$.*

We provide an outline of the proof, with a detailed derivation available in Appendix 8. The optimal value function $V(\mathbf{x}_t)$ of the optimal control problem defined in equation 7 satisfies the Bellman optimality equation,

$$V(\mathbf{x}_t) = \mathcal{L}(\{\mathbf{x}_s\}_{s=0}^t, \pi_t, (\mathbf{Q}_t^P, \mathbf{Q}_t^I, \mathbf{Q}_t^D)) + V(\mathbf{x}_{t+1}), \quad \text{s.t. } \mathbf{x}_{t+1} = \boldsymbol{\theta}_t(\mathbf{x}_t + \pi_t(\mathbf{x}_t)).$$

In the linear case, the optimal value function is parametrized by a quadratic function, expressed as $V(\mathbf{x}_t) = \mathbf{x}_t^\top \mathbf{P}_t \mathbf{x}_t$. By setting the derivative $\frac{dV(\mathbf{x}_t)}{d\pi_t(\mathbf{x}_t)}$ to zero, we arrive at the optimal control solution, as detailed in equation 9. Furthermore, the Riccati equation in equation 8 emerges from substituting this optimal control solution into the Bellman optimality equation.

**Remark 2.** *As derived in equation 9, using a combination of P, I, and D controllers incurs the same computational cost as using a single type of control scheme. This is due to the linearity of the control process, where the orthogonal projections onto the state embedding, state integration embedding, and state derivative embeddings can be effectively merged. This results in a projection onto the intersecting subspace of the three linear embedding subspaces.*

Starting with a pre-trained LLM, the layer-wise transformations can be linearized to form a linear dynamical system. From this, the parameters of the optimal value function $\mathbf{P}_t$ are computed using the discrete dynamical system outlined in equation 8. Subsequently, the optimal feedback control solution $\pi_t(\mathbf{x}_t)$ is constructed from equation 9. Although this method is feasible, the linearization of a series of transformer layers poses its own set of complexities. Moving forward, we propose an analytic solution that does not rely on linearizing the base model, under additional orthogonality assumptions.

**An analytic solution under Assumption** 2. We further consider the scenario where both embedding manifold and layer-wise transformation are orthogonal. As a result, the linear combination of orthogonal projections, represented as $\mathbf{Q}_t = \mathbf{Q}_t^P + \mathbf{Q}_t^I + \mathbf{Q}_t^D$, forms an orthogonal projection itself. With these conditions in place, we can then establish an analytic formulation for the optimal control solution, as detailed in the following proposition.

**Proposition 3.** *When the layer-wise transformations are represented as orthogonal matrices, and the basis of state embedding, state integration embedding, and state derivative embeddings are mutually orthogonal, the optimal feedback control, denoted as $\pi_t(\mathbf{x}_t)$, can be computed as follows:*

$$\pi_t(\mathbf{x}_t) = -\mathbf{V}_t \begin{bmatrix} 0 & 0 & \cdots & 0 & 0 \\ 0 & 0 & \cdots & 0 & 0 \\ \vdots & \vdots & \ddots & 0 & 0 \\ 0 & 0 & \cdots & 1 - \frac{c}{1+\lambda_{t+1}+c} & 0 \\ 0 & 0 & \cdots & 0 & 1 - \frac{c}{1+\lambda_{t+1}+c} \end{bmatrix} \mathbf{V}_t^\top \mathbf{x}_t,$$

*where the time-varying parameter $\lambda_t$ is governed by a backward difference equation $\lambda_t = \frac{c(1+\lambda_{t+1})}{1+\lambda_{t+1}+c}$, with the terminal condition specified as $\lambda_T = 0$.*

A brief overview of the proof is presented here, while a more detailed derivation can be found in Appendix 8. The condition for the embedding subspaces to be orthogonal guarantees that the linear combination represented by $\mathbf{Q}_t = \mathbf{Q}_t^P + \mathbf{Q}_t^I + \mathbf{Q}_t^D$ forms an orthogonal projection. Moreover, the orthogonality in layer-wise transformations simplifies the Riccati equation. This simplification leads to a recursive approach to formulating control regularization.

When $c = 0$, it holds that $\lambda_t = 0$ for every $t$, and the optimal feedback control corresponds to the orthogonal projection onto the orthogonal complement of the linear subspace. On the other hand, for $c > 0$, the approach yields a time-varying regularization in control across different layers. This analytical solution, which assumes linear orthogonality, is independent of the underlying model. Therefore, the time-variant control regularization $c_t$ can be pre-calculated prior to the inference process.

## 2.3 Theoretical Error Analysis

Under both assumptions 1 and 2 defined in Section 2.2, the controlled dynamics, under some perturbations, are formulated as follows:

$$\bar{\mathbf{x}}_{t+1} = \boldsymbol{\theta}_t(\bar{\mathbf{x}}_t + \pi_t(\bar{\mathbf{x}}_t)), \ \ \bar{\mathbf{x}}_0 = \mathbf{x}_0 + \mathbf{z},$$

where $\mathbf{z}$ represents an arbitrary perturbation decomposable into two mutually orthogonal components $\mathbf{z} = \mathbf{z}^{\parallel} \oplus \mathbf{z}^{\perp}$: $\mathbf{z}^{\parallel}$, aligning within the data embedding subspace, and $\mathbf{z}^{\perp}$, orthogonal to the data manifold. We represent the state trajectory in the absence of input perturbation and without any applied control as $\mathbf{x}_{t+1} = \boldsymbol{\theta}_t(\mathbf{x}_t)$, the following theorem quantifies the error as $\|\bar{\mathbf{x}}_t - \mathbf{x}_t\|_2^2$, which evaluates the difference between the perturbed state after control correction and the original state from unperturbed input data.

**Theorem 4.** *For any time step $t \geq 1$, assuming that each $\boldsymbol{\theta}_t$ is an orthogonal matrix, we have the following error computation:*

$$\|\bar{\mathbf{x}}_t - \mathbf{x}_t\|_2^2 = \prod_{s=0}^{t-1} \alpha_s^2 \cdot \|\mathbf{z}^{\perp}\|_2^2 + \|\mathbf{z}^{\parallel}\|_2^2,$$

*where $\alpha_t$ is a time-varying parameter defined in relation to the control regularization $c$, and $\lambda_t$ are auxiliary variables, as follows:*

$$\alpha_t = \frac{c}{1 + \lambda_{t+1} + c}, \quad \lambda_T = 0, \quad \lambda_{T-1} = \frac{c}{1+c}, \quad \lambda_t = \frac{c(1 + \lambda_{t+1})}{1 + c + \lambda_{t+1}}.$$

The detailed derivation is provided in Appendix 9. This computation rigorously demonstrates that perturbations, specifically those spanning the orthogonal complement denoted by $\mathbf{z}^{\perp}$, exhibit a decay phenomenon during the process of forward propagation. Furthermore, in scenarios where control parameters are subject to regularization constraints, when $c > 0$, our analysis reveals nuanced insights. We establish that the optimal control solution, which is derived by considering the intricate interplay among different transformation layers, adheres to these constraints while optimizing performance, which captures the complex dynamics between layers.

Theorem 4 outlines how errors in state computations at any given time step are influenced by input perturbations represented by $\mathbf{z}$, despite these perturbations existing within the continuous domain of $\mathbb{R}^d$, this setting fits real-world adversarial attacks on LLMs, which involve modifying discrete elements, such as tokens or characters, in the input text. The act of modifying a word or substring through an adversarial attack leads to a discrepancy between the embedding sequences of the original and modified input tokens, manifesting as the perturbation vector $\mathbf{z}$ within the input embedding space. Specifically, the embedding manifolds, derived from unperturbed training data, capture the structure of this data in a lower-dimensional subspace. Adversarial examples, meanwhile, are designed to be semantically similar to the original input yet induce a marked divergence in the embedding space during the model's forward propagation. Under these circumstances, the difference between the embedding sequences of the original input and the adversarial example can be quantified and adjusted within the PID control framework. We provide empirical error computation of Theorem 4 in Section 3.4.

---

**Algorithm 1** Tucker Decomposition.

---

**Input:** An $I$-way tensor $\boldsymbol{\mathcal{X}}$.
**Output:** Core tensor $\boldsymbol{\mathcal{G}}$, orthogonal basis $\mathbf{V}^1, \mathbf{V}^2, \cdots, \mathbf{V}^I$.
**for** $i = 1$ to $I$ **do**
    $\mathbf{X}^i = $ Reshape $(\boldsymbol{\mathcal{X}}, i)$, // Reshape the tensor along the i$^{\text{th}}$ mode.
    $\mathbf{U}^i, \mathbf{S}^i, \mathbf{V}^i = $ SVD $(\mathbf{X}^i)$, // Perform singular value decomposition on the reshaped tensor.
    Save the singular vectors $\mathbf{V}^i$ as the orthogonal basis.
**end for**
$\boldsymbol{\mathcal{G}} = \boldsymbol{\mathcal{X}}$, // Initialize the tensor core with the $I$-way tensor $\boldsymbol{\mathcal{X}}$.
**for** $i = 1$ to $I$ **do**
    $\boldsymbol{\mathcal{G}} = \boldsymbol{\mathcal{G}} \times_i \mathbf{V}^i$, // Multiply the core tensor by the i$^{\text{th}}$ orthogonal basis.
**end for**

---

### 2.4 Additional Details for Constructing PID Control

Here, we provide details on implementing the proposed PID control method. We begin with constructing the linear embedding basis $\mathbf{V}_t^P$, $\mathbf{V}_t^I$, and $\mathbf{V}_t^D$ from training dataset. In NLP tasks, the hidden states are generally represented as 2-dimensional matrix (sequence of embedding vectors), $\mathbf{X}_t \in \mathbb{R}^{l \times d}$, where $l$ denotes the temporal length. Given $N$ pieces of data sampled from the data distribution $\mathcal{D}$ ($N$ training data), we can concatenate the hidden states as a 3-way tensor $\boldsymbol{\mathcal{X}}_t \in \mathbb{R}^{N \times l \times d}$, and apply Tucker decomposition (De Lathauwer et al., 2000b;a; Kolda & Bader, 2009) (known as high-order singular value decomposition) to generate linear embedding basis along both temporal and state embedding dimensions.

Tucker Decomposition is an extension of the traditional singular value decomposition to higher-order tensors. Mathematically, Tucker decomposition represents an $I$-way tensor as $\boldsymbol{\mathcal{X}} \approx \boldsymbol{\mathcal{G}} \times_1 \mathbf{V}^{(1)} \times_2 \mathbf{V}^{(2)} \times_3 \cdots \times_I \mathbf{V}^{(I)}$, where $\boldsymbol{\mathcal{G}}$ is the core tensor, which governs the interaction between different modes, $\mathbf{V}^i$ are orthogonal bases corresponding to the principal components in each tensor mode, $\times_n$ is the mode-$n$ tensor product. The mode-$n$ product of a tensor $\boldsymbol{\mathcal{A}} \in \mathbb{R}^{I_1 \times \cdots \times I_n \times \cdots \times I_d}$ with a matrix $\mathbf{U} \in \mathbb{R}^{J \times I_n}$ is defined as

$$\boldsymbol{\mathcal{B}} = \boldsymbol{\mathcal{A}} \times_n \mathbf{U} \iff b_{i_1 \ldots i_{n-1} j i_{n+1} \ldots i_d} = \sum_{i_n=1}^{I_n} a_{i_1 \ldots i_n \ldots i_d} \cdot u_{j i_n},$$

where the $(i_1, i_2, \cdots, i_d)$-th elements of $\boldsymbol{\mathcal{A}}$ and $\boldsymbol{\mathcal{B}}$ are denoted as $a_{i_1 i_2 \cdots i_d}$ and $b_{i_1 i_2 \cdots i_d}$, respectively.

An implementation of Tucker decomposition is detailed in Algorithm 1. Along each of the $I$ modes, the concatenated high-dimensional states $\boldsymbol{\mathcal{X}}$ are reshaped along the i$^{\text{th}}$ dimension, which is used to compute the orthogonal basis from singular value decomposition. The core tensor $\boldsymbol{\mathcal{G}}$ is computed by multiplying the states $\boldsymbol{\mathcal{X}}$ with each of the $I$ basis along each mode. The low-rank reconstruction of concatenated states $\boldsymbol{\mathcal{X}}$ can be obtained by $\boldsymbol{\mathcal{G}} \times_1 \mathbf{V}^1 \times_2 \mathbf{V}^2 \times_3 \cdots \times_I \mathbf{V}^I$.

Given a pre-trained LLM (naively trained or robustly trained), we collect the concatenated states from training data, which results in a set of 3-way tensors $\{\boldsymbol{\mathcal{X}}_t\}_{t=0}^{T-1}$. Then Tucker decomposition is applied at every $\boldsymbol{\mathcal{X}}_t$ (refer Algorithm 1). Extending this to integral and derivative controls is straightforward, as one can substitute the concatenated states $\boldsymbol{\mathcal{X}}$ by either the summation of past states $\boldsymbol{\mathcal{X}} = \sum_{s=0}^t \boldsymbol{\mathcal{X}}_s$ or the subtraction of two consequential states $\boldsymbol{\mathcal{X}} = \boldsymbol{\mathcal{X}}_t - \boldsymbol{\mathcal{X}}_{t-1}$. Using the linear embedding bases $\mathbf{V}_t^P$, $\mathbf{V}_t^I$, and $\mathbf{V}_t^D$ obtained from Tucker decomposition, the construction of the feedback controller is achieved by adhering to the methodology outlined in Proposition 3.

## 3 Numerical Experiments

In this section, we first discuss experimental setup in Section 3.1. In Sections 3.2 and 3.3, we assess the performance of the proposed PID control framework against various baseline methods across multiple NLP tasks. Subsequently, in Section 3.4, an ablation study is conducted, providing exploratory justification for the proposed approach.

### 3.1 Experimental Setup

**Evaluation methods:** We consider both adversarial attack algorithms (e.g. A2T, PSO, TextBugger, TextFooler), applied on the SNLI (Bowman et al., 2015), MNLI datasets (Williams et al., 2018) and adversarial datasets (e.g. ANLI) to evaluate the robustness of the proposed PID control and baselines.

- **A2T** (Attacking to Training (Yoo & Qi, 2021)) utilizes a cost-effective gradient-based technique to rank the significance of words. This approach encompasses the iterative replacement of each word with synonyms sourced from counter-fitted word embeddings.

- **PSO** (Zang et al., 2020) exploits a population of interacting individuals to iteratively search for the optimal solution in the specific space.

- **TextBugger** (Li et al., 2019) finds important words by computing the Jacobian matrix of the model and then chooses an optimal perturbation from the generated perturbations.

- **TextFooler** (Jin et al., 2020) is the state-of-the-art word-level adversarial attack method to generate adversarial examples. This technique identifies the important words for the target model and subsequently prioritizes their replacement with the most semantically similar and grammatically correct words. This process continues until there is a discernible shift in the model's prediction.

- Adversarial NLI (**ANLI**) (Nie et al., 2020) is a large-scale NLI benchmark, This dataset was curated through an iterative process that incorporates both human and model inputs in an adversarial loop, targeting specific models for attack. The ANLI dataset is particularly potent as an adversarial tool, demonstrating a significant capability to diminish the accuracy of pre-trained models.

**Baseline methods:** This study examines two baseline methods focused on adversarial training. The Naive adversarial training (**AT**), as proposed by Yoo & Qi (2021), employs the A2T attack for its adversarial training process. **FreeLB**, introduced by Zhu et al. (2019), implements adversarial training in language models during the fine-tuning stage, aiming to enhance both generalization and robustness. It is noteworthy that the PID control method, in contrast to these adversarial training-based approaches, offers a distinct perspective on enhancing model robustness without knowing the attack type in advance. It can be applied to models that have undergone adversarial training to further improve their robustness.

We fine-tune four baseline models using LoRA (Hu et al., 2021), namely **distilbert** (Sanh et al., 2019), **BERT-large** (Kenton & Toutanova, 2019), **RoBERTaBase** and **RoBERTaLarge** (Liu et al., 2019).

**PID control implementation details:** Using a pre-trained model (e.g., BERT), we select training data that this model can accurately predict. Next, we simulate forward propagation using the pre-trained model on this specific set of training data, which generates a collection of 3-dimensional tensors, denoted as $\{\mathcal{X}_t\}_{t=0}^{T-1}$. Following this, we employ Algorithm 1 on each tensor to determine the basis for a linear embedding subspace (see Section 2.4). The dimension of this subspace is chosen based on the criterion that it must account for 99% of the total variance observed (this is done by accumulating the singular values). Finally, the optimal solution outlined in Proposition 3 is implemented to generate a time-dependent control regularization parameter.

**Threat model:** In this work, we consider word/token level adversarial attacks that manipulate discrete tokens or characters in the input text. We verify this from both empirical and theoretical perspectives. In the numerical experiments, we consider a range of adversarial attacks, including A2T, PSO, TextBugger, and TextFooler. These adversarial attacks aim to cause misclassification by modifying the tokens of the input string while maintaining the same semantic meaning.

### 3.2 Robustness Against Adversarial Examples

Here, we empirically validate the robustness improvement of employing the proposed PID control on pre-trained LLMs. In Figure 2 (a), a radar plot is presented to illustrate the comparative performance between the baseline and controlled models, utilizing the DistilBERT architecture and evaluated on the SNLI dataset.

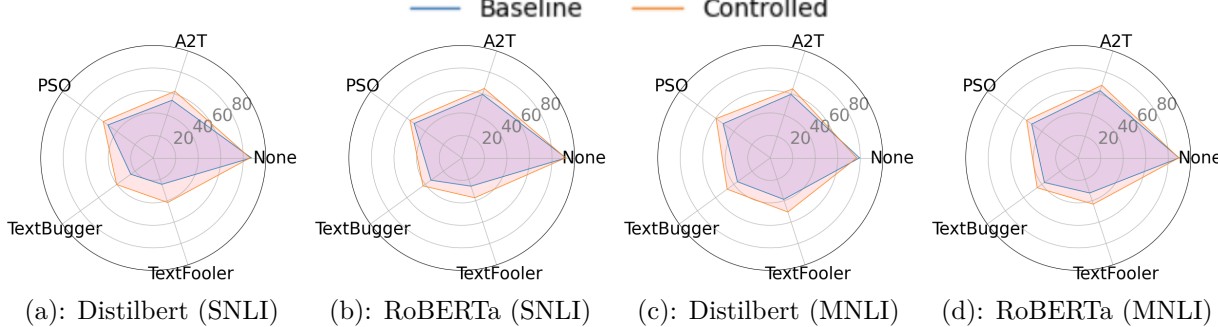

(a): Distilbert (SNLI)   (b): RoBERTa (SNLI)   (c): Distilbert (MNLI)   (d): RoBERTa (MNLI)

Figure 2: (a) and (b) are radar plots that summarize Distilbert and RoBERTaLarge in Table 8 for SNLI dataset, respectively. (c) and (d) are radar plots that summarize Distilbert and RoBERTaLarge in Table 9 for MNLI dataset, respectively.

This demonstrates that the employment of PID control significantly improves model robustness against all four distinct types of perturbations, with a negligible impact on performance with unaltered data (denoted as None). Shifting to a different model architecture, Figure 2 (b) reveals that applying the proposed PID control approach to the RoBERTa model yields analogous enhancements in robustness. For more challenging scenarios, Figures 2 (c) and (d) detail the performance of the MNLI dataset. In these plots, both the DistilBERT and RoBERTa architectures are examined. These figures showcase that, despite the increased complexity of the MNLI dataset, the PID control method consistently maintains robustness improvements. The increased complexity of the MNLI dataset poses additional challenges in creating embedding subspaces, making it more difficult to accurately represent state, state integration, and state derivatives with linear embeddings. The plots distinctly highlight that the controlled models exhibit increased resistance to a broader spectrum of linguistic perturbations and complexities, without significant trade-offs in overall accuracy. This underlines the efficacy of PID control in enhancing model robustness across different architectures and datasets.

More detailed comparisons of performance between baseline and controlled models on the SNLI and MNLI datasets are provided in Tables 8 and 9 in Appendix 10. When a method's accuracy surpasses others by more than 1%, it's highlighted in red. It is evident that the proposed PID control method significantly enhances the robustness of both standard and robustly trained LLMs. The enhancement is more pronounced in standard trained models, which are generally more vulnerable to adversarial attacks. On average, the PID control method yields an improvement of nearly 10% in standard models and about 5% in robustly trained models, including both AT and FreeLB training.

In addition, we present the numerical results of OPT-1.3B. OPT-1.3B is a decoder-based large language model that contains 1.3 billion model parameters. For the proposed PID control, we follow the same P-D control implementation (proportional-derivative) as done in all numerical experiments from the paper. Table 1 demonstrates that the controlled OPT-1.3B model consistently improves the robustness performance against all four types of adversarial attacks. Specifically, on the SNLI dataset, the average improvement is over 20% compared with the base model. On a more challenging MNLI dataset, with only a 2.5% accuracy drop on the unperturbed testing dataset. The improvement reaches 21% against the TextBugger attack, and 11% on both A2T and PSO attacks.

Table 1: Measurement on SNLI dataset: baseline model / controlled model

| | None | A2T | PSO | TextBugger | TextFooler |
|---|---|---|---|---|---|
| SNLI Dataset | | | | | |
| OPT | 91.24 / 88.69 | 49.15 / 63.28 | 48.00 / 60.06 | 17.57 / 41.79 | 16.64 / 44.70 |
| MNLI Dataset | | | | | |
| OPT | 86.89 / 84.27 | 54.47 / 65.87 | 45.14 / 59.08 | 24.12 / 45.97 | 21.68 / 49.13 |

Table 2: Measurement on ANLI dataset: baseline model / controlled model

|  | r1 | r2 | r3 |
|---|---|---|---|
| RoBERTaLarge (Dev) | 72.60 / 72.65 | 50.99 / 52.33 | 40.99 / 43.31 |
| RoBERTaLarge (Test) | 72.79 / 72.60 | 48.19 / 49.39 | 40.66 / 42.41 |

### 3.3 Robustness Against Adversarial Dataset

In this study, we assess the effectiveness of the PID control approach in an adversarial Natural Language Inference (NLI) task. The ANLI dataset is created through an iterative process involving both humans and models, aimed at improving natural language understanding. Initially, human annotators create examples that challenge the current best-performing models. These difficult examples, intended to reveal more weaknesses, are then incorporated into the training set to enhance the models. This cycle of identifying and addressing weaknesses is repeated across several rounds, each producing an increasingly complex adversarial dataset (ANLI consists of three rounds of development and test datasets). Unlike the evaluation using adversarial examples described in Section 3.2, the ANLI dataset is pre-constructed by human annotators. In contrast, adversarial examples from Section 3.2 are created in relation to the specific characteristics of the underlying classifier.

The evaluation with the ANLI dataset encompasses both baseline and controlled models, utilizing the development and test datasets. The results obtained from the ANLI dataset are outlined in Table 2. ANLI involves three progressively challenging rounds. The baseline model shows a decline in performance with increasing difficulty from round 1 to round 3. Conversely, the PID control demonstrates a more pronounced improvement in performance as the challenge increases. Specifically, the proposed control method leads to 1.0783% in the mean of performance improvement, and a 95% confidence interval of 0.0564% to 2.1004%.

### 3.4 Ablation Study

This section provides exploratory justifications for the proposed PID control framework.

**Justification of the selection of a P-D control scheme.** We begin with a comparative analysis of various control schemes, emphasizing the benefits of implementing multiple controllers over the single use of Proportional (P) control as previously explored in (Chen et al., 2020). Table 3 showcases a comparison of the robustness performance across Proportional (P), Proportional-Integral (P-I), Proportional-Derivative (P-D), and Proportional-Integral-Derivative (P-I-D) control schemes within different model architectures and training methodologies. It is evident that the P-D control scheme significantly surpasses the others in most scenarios, underscoring the efficacy of the proposed PID control framework, which expands upon the limited capability of earlier P control (closed-loop control) methods. The mean of employing the Proportional-Derivative (P-D) control over the Proportional (P) control is 2.35%, with a 95% confidence interval of 1.677% to 3.0121%. This validates the choice of P-D control.

The reason why P-D outperforms P-I-D is mainly due to noise sensitivity and hyperparameter tuning.

Noise sensitivity: The integral term has the potential to aggregate errors across multiple hidden layers, incorporating noise inherent in the embedding manifolds, as well as the distributional shift between the training and testing datasets. In scenarios where substantial noises are presented in each hidden layer, the integral component, dependent on the embedding manifold of accumulated past states, may lead to instability during model inference. Conversely, a Proportional-Derivative (PD) controller, lacking the integral component, tends to exhibit improved performance under such noisy conditions by not accumulating this noise.

Hyperparameter tuning: In the realm of traditional PID control design, selecting the appropriate control gains, denoted as $K_p$, $K_i$, and $K_d$, for proportional, integral, and derivative controls respectively, presents a notable challenge. These gains are crucial for achieving a balance among the different types of controls. Typically, the calibration of these gains is empirically based, with the aim of optimizing the performance of

Table 3: Measurement on SNLI dataset: P / P-I / P-D / P-I-D

| Distilbert | | | |
|---|---|---|---|
| | Standard training | Adversarial training | FreeLB |
| A2T | 60.11 / 58.24 / 62.31 / 61.30 | 72.09 / 71.12 / 71.81 / 71.97 | 59.63 / 57.90 / 62.95 / 61.88 |
| PSO | 53.39 / 52.67 / 54.96 / 53.96 | 55.80 / 54.72 / 57.87 / 56.35 | 54.40 / 53.91 / 56.86 / 55.89 |
| TextBugger | 37.15 / 37.15 / 40.26 / 39.54 | 38.91 / 38.98 / 41.64 / 40.98 | 32.79 / 32.24 / 37.80 / 36.21 |
| TextFooler | 36.81 / 34.84 / 41.73 / 38.98 | 40.15 / 38.21 / 43.81 / 41.12 | 32.49 / 31.22 / 39.64 / 37.39 |

| RoBERTaBase | | | |
|---|---|---|---|
| | Standard training | Adversarial training | FreeLB |
| A2T | 61.78 / 60.81 / 64.11 / 61.94 | 76.63 / 75.82 / 77.08 / 76.28 | 65.93 / 65.04 / 68.85 / 66.59 |
| PSO | 53.34 / 52.94 / 54.40 / 53.35 | 55.49 / 54.76 / 56.45 / 54.99 | 53.64 / 52.99 / 55.24 / 53.55 |
| TextBugger | 40.46 / 39.00 / 43.20 / 40.53 | 41.71 / 40.22 / 43.35 / 41.33 | 39.72 / 38.24 / 42.75 / 40.17 |
| TextFooler | 33.19 / 32.10 / 37.35 / 33.83 | 34.48 / 32.47 / 39.39 / 35.29 | 30.75 / 29.77 / 36.81 / 32.21 |

| BERT-large | | | |
|---|---|---|---|
| | Standard training | Adversarial training | FreeLB |
| A2T | 75.75 / 75.68 / 75.54 / 75.60 | 86.13 / 86.03 / 85.76 / 85.92 | 78.16 / 78.16 / 78.21 / 78.04 |
| PSO | 67.72 / 67.69 / 67.55 / 67.60 | 70.21 / 70.26 / 70.38 / 70.25 | 65.49 / 65.46 / 65.56 / 65.46 |
| TextBugger | 64.59 / 64.53 / 64.41 / 64.36 | 69.74 / 69.89 / 69.55 / 69.62 | 59.28 / 59.35 / 59.29 / 59.27 |
| TextFooler | 58.48 / 58.25 / 58.27 / 58.12 | 65.43 / 65.25 / 65.27 / 65.10 | 55.34 / 55.27 / 55.26 / 55.14 |

| RoBERTaLarge | | | |
|---|---|---|---|
| | Standard training | Adversarial training | FreeLB |
| A2T | 65.10 / 64.89 / 64.95 / 64.38 | 81.91 / 81.72 / 81.62 / 81.80 | 70.38 / 70.40 / 71.30 / 70.51 |
| PSO | 55.83 / 55.04 / 56.70 / 55.31 | 57.99 / 57.29 / 59.71 / 58.18 | 56.23 / 55.60 / 57.20 / 56.18 |
| TextBugger | 44.61 / 42.20 / 42.43 / 41.29 | 45.00 / 43.54 / 44.74 / 43.53 | 44.52 / 43.11 / 44.42 / 43.21 |
| TextFooler | 36.63 / 35.52 / 37.29 / 35.39 | 39.64 / 37.06 / 42.44 / 39.87 | 37.56 / 35.97 / 38.59 / 36.71 |

PID control. Our method follows a similar strategy, determining the gains through experimentation with training data. Given that our hyperparameter searching space only contains 0 and 0.5 for each control gain, this results in the values $K_p = 0.5$, $K_d = 0.5$, and $K_I = 0$. A more principled method would entail adjusting these hyper-parameters through numerical optimization, treating these control gains as adjustable variables. The development of a more sophisticated strategy for fine-tuning the control gains will be studied for future exploration.

**Discussion on the linearity and orthogonality assumptions.** Here we discuss the negative impact of violating the assumptions made to derive the analytic solution. Through empirical evaluations, we highlight how the main assumptions have increasingly adverse effects, especially when the embedding manifolds fail to accurately capture the complex, high-dimensional states. More specifically, applying regularization on control solutions can mitigate these inaccuracies. However, as the precision of the embedding manifolds decreases, a greater degree of regularization is required, thereby complicating the optimal control problems. The increased complexity in the optimal control problem makes the negative impact of violating the main assumptions more significant.

Our validation approach involves a performance comparison between the proposed analytic solution and the implementation of Pontryagin's Maximum Principle, an iterative solver that operates without the need for additional assumptions. Pontryagin's Maximum Principle provides the necessary conditions for an optimal control solution, typically offering a robust approximation of such solutions. We further elaborate this comparison by creating linear embedding subspaces with varying thresholds for accumulated variances, specifically aiming to capture 99%, 95%, 90%, and 85% of the variances in the underlying states. As the variance threshold is lowered, the accuracy of these embedding subspaces decreases, thus posing greater challenges in solving optimal control problems. The performance comparison, detailed in Table 4, includes three LLMs across five evaluation tasks. These tasks include a standard scenario with no perturbation

Table 4: Performance Comparison (Analytic Solution / PMP)

| | Base | 0.99% | 0.95% | 0.9% | 0.85% |
|---|---|---|---|---|---|
| Distilbert | | | | | |
| None | 87.23 | 85.88 / 86.52 | 68.92 / 80.21 | 34.24 / 54.06 | 34.28 / 46.75 |
| A2T | 53.89 | 61.75 / 60.87 | 57.93 / 62.88 | 34.10 / 44.17 | 34.28 / 42.01 |
| PSO | 49.84 | 54.33 / 52.80 | 56.21 / 58.22 | 34.13 / 46.27 | 34.28 / 42.55 |
| TextBugger | 24.73 | 40.35 / 36.69 | 43.89 / 42.50 | 36.24 / 39.02 | 34.28 / 42.20 |
| TextFooler | 24.69 | 40.28 / 36.13 | 49.05 / 46.69 | 34.37 / 39.56 | 34.28 / 40.80 |
| RoBERTaBase | | | | | |
| None | 90.87 | 90.10 / 90.59 | 85.09 / 89.63 | 64.60 / 85.82 | 40.02 / 76.71 |
| A2T | 58.36 | 63.82 / 62.19 | 65.12 / 66.44 | 51.19 / 66.86 | 37.56 / 60.41 |
| PSO | 51.44 | 54.36 / 52.98 | 59.97 / 56.75 | 52.55 / 59.18 | 37.91 / 59.07 |
| TextBugger | 35.90 | 43.03 / 40.53 | 46.74 / 46.41 | 38.72 / 47.36 | 34.84 / 42.43 |
| TextFooler | 27.03 | 37.18 / 33.47 | 47.16 / 42.79 | 41.40 / 46.60 | 35.76 / 45.49 |
| RoBERTaLarge | | | | | |
| None | 92.39 | 91.98 / 92.11 | 86.50 / 91.68 | 66.40 / 90.53 | 44.54 / 86.52 |
| A2T | 59.40 | 64.64 / 63.15 | 67.17 / 65.03 | 54.67 / 64.99 | 41.54 / 61.94 |
| PSO | 52.15 | 56.62 / 54.35 | 62.14 / 55.51 | 55.13 / 58.41 | 41.15 / 58.85 |
| TextBugger | 33.72 | 42.39 / 39.18 | 47.48 / 41.59 | 41.30 / 43.77 | 35.49 / 40.32 |
| TextFooler | 26.43 | 36.92 / 32.27 | 48.79 / 37.14 | 47.09 / 41.43 | 40.47 / 41.97 |

and four adversarial attacks: A2T, PSO, TextBugger, and TextFooler. The results reveal that while the performance difference between the analytic solution and Pontryagin's Maximum Principle is negligible at higher accuracy levels (e.g., 99% variance), the scenario changes significantly at lower accuracies (e.g., 90% and 85% variances). In these instances, employing Pontryagin's Maximum Principle, which operates independently of simplifying assumptions, yields noticeably better control solutions.

**Computational wall time comparison.** Here we present a detailed discussion of the computation overhead of the proposed PID control method. Specifically, we compare the computational wall time between the base model without any controls applied, the proposed analytic solution, and Pontryagin's maximum principle employed in the previous closed-loop control approach. As shown in Table 5, across all four models, the computational wall time between the base model and the proposed analytic solution is comparable, the analytic solution only adds a small amount wall time during inference. However, solving the PMP significantly adds the computational wall time of the base model.

Table 5: Computational Wall Time

| Wall Time (s) of $10,000$ Test Samples (averaged over 5 experiments) | | | | |
|---|---|---|---|---|
| | Distilbert | RoBERTaBase | RoBERTaLarge | OPT |
| Base model | 6.3751 | 11.7756 | 36.0178 | 123.5379 |
| Controlled model | 6.4221 | 11.8620 | 36.5051 | 124.1090 |
| PMP | 62.2667 | 81.2795 | 263.7920 | 757.0649 |

**Error computation of the main theorem.** We provide the details of the error computation outlined in Theorem 4. Our objective is to demonstrate that the accuracy of the error computation specified in Theorem 4 diminishes with the addition of more layers to the language model. This decrease in accuracy is due to the assumptions of linearity and orthogonality. According to these assumptions, the transformations applied to each layer of a language model merely rotate the hidden state without altering its magnitude. However, as

the model incorporates more of these layer-wise transformations, the accuracy of these assumptions starts to decrease.

Table 6 presents the calculation of the absolute difference between the actual error and the error estimate as per Theorem 4. It is evident that with all types of adversarial perturbations (A2T, PSO, TextBugger, and TextFooler), the increase in the number of layers within the language model (with 6 layers representing Distilbert, 12 layers symbolizing RoBERTaBase, and 24 layers signifying RoBERTaLarge) leads to a rise in the absolute error. This indicates a decline in the precision of the error estimation.

Table 6: Error Comparison (difference between Theorem 4 and true error)

|  | 6 Layers | 12 Layers | 24 Layers |
|---|---|---|---|
| A2T | 3.2189 | 4.3062 | 5.1566 |
| PSO | 1.9156 | 2.6087 | 3.3047 |
| TextBugger | 3.2189 | 4.3062 | 5.1566 |
| TextFooler | 3.1348 | 4.2894 | 5.2915 |

**Justification on the effectiveness of the PID control framework.** Here we provide evidence that the PID control framework can improve model robustness against adversarial examples. As detailed in Theorem 4, the working principle of the PID control framework is based on the two facts:

- There exists an embedding structure in the state at every layer. Table 7 verifies the existence of lower-dimensional embedding subspaces. With the OPT-1.3 B LLM (only the first 6 layers are shown), the dimensions of proportional, integral, and derivative embedding subspaces are presented. As can be seen, the dimension of proportional embedding is around 350 on average in a 2048 dimensional space, integral and derivative embedding subspaces also show low dimensions compared with the full space.

- The sequence of states from adversarially perturbed input deviates from the true embedding structures. Table 7 presents the error (measured in 2-norm) detected by the combination of P, I, and D embedding subspaces. As can be seen, the perturbation aims to amplify the error as propagated into deeper layers, and the embedding subspaces can effectively detect these errors at all layers.

Table 7: Ranks and Embedding Errors

| Ranks: OPT-1.3B (first 6 layers) | | | | | | |
|---|---|---|---|---|---|---|
|  | Layer 1 | Layer 2 | Layer 3 | Layer 4 | Layer 5 | Layer 6 |
| Proportional | 191 / 2048 | 148 / 2048 | 224 / 2048 | 337 / 2048 | 451 / 2048 | 549 / 2048 |
| Integral | 191 / 2048 | 181 / 2048 | 180 / 2048 | 212 / 2048 | 253 / 2048 | 298 / 2048 |
| Derivative | 191 / 2048 | 355 / 2048 | 1001 / 2048 | 1352 / 2048 | 1494 / 2048 | 1535 / 2048 |
| Embedding error: OPT-1.3B (first 6 layers) | | | | | | |
| OPT | 1.8237 | 9.5877 | 6.9810 | 6.3207 | 7.5278 | 16.6526 |

## 4 Related Works

We delve into the existing body of literature surrounding robustness issues in NLP tasks (Section 4.1). Moreover, we explore the realm of machine learning from an optimal control perspective, emphasizing its relevance and applicability to the task at hand (Section 4.2).

### 4.1 Robustness in NLP

In recent years, a variety of approaches have been developed for generating effective adversarial attacks in the context of NLP. Traditionally, text attacks are produced through direct modifications to sentences at the

character level, word level, sentence level, or a combination of these Ren et al. (2019); Li et al. (2018a); Xu et al. (2021). Studies such as those by Liu et al. (2020b); Bělohlávek et al. (2018); Zhu et al. (2019) explore methods for generating attacks within the text embedding space. An alternative approach by Wallace et al. (2019) involves prepending the same tokens to all texts for attacks. Moving away from adding perturbations to samples, La Malfa & Kwiatkowska (2022); Wang et al. (2020) integrate generative models for the creation of attacks.

In response to adversarial attacks, most defense strategies primarily rely on the principles of adversarial training Morris et al. (2020); Jiang et al. (2019); Wu et al. (2022). Given that adversarial training is inherently resource-intensive, both in terms of computation and time, studies such as Zhu et al. (2019); Zhang et al. (2019); Shafahi et al. (2019) have proposed methods to expedite the training process. However, despite the efficacy of adversarial training, there's evidence suggesting that fine-tuning model parameters can diminish performance on clean data He et al. (2021). Additionally, it has been observed that adversarial training often produces suboptimal results when faced with novel, unforeseen perturbations Tramer & Boneh (2019). Adversarial training is analogous to open-loop control, as it leverages a set of fixed controls (e.g. model parameters) for any new, unobserved data. In contrast, the proposed PID control approach dynamically focuses on inputs using feedback controls.

## 4.2 Deep Learning with Optimal Control

Recent studies have increasingly focused on elucidating the intricate connections between dynamical systems and deep learning, highlighting the fundamental interrelations that exist between them (E, 2017; Haber & Ruthotto, 2017; Li et al., 2018b). These studies have established a robust theoretical framework, offering a novel perspective to comprehend deep learning methodologies through the lens of optimal control theory (Liu & Theodorou, 2019). The work of Li et al. (2018b) and Li & Hao (2018) have successfully bridged the gap between the classical back-propagation algorithm and optimal control theory. This intersection notably highlights how Pontryagin's Maximum Principle (Kirk, 1970), a cornerstone of control theory, aligns closely with gradient-based training methods in neural networks.

In advancing this line of inquiry, E et al. (2018) has developed the theoretical foundations for interpreting deep learning within the optimal control framework. These insights have laid the groundwork for subsequent research that applies optimal control principles to address key challenges in deep learning. A notable example of this is the work by (Liu et al., 2020a), which introduced sophisticated high-order optimization strategies rooted in differential dynamic programming. This methodology has been instrumental in enhancing the training process's convergence rates and stability. Moreover, the closed-loop control framework has been proposed to improve the model's robustness against adversarial attacks (Chen et al., 2020; 2022) and fairness issues (Chen et al., 2023).

## 5 Discussion and Future Works

**The wider perspective of the proposed method on the trustworthy ML:** The presented PID control approach generalizes previous closed-loop control approaches with additional integral and derivative controllers. This development leads to more flexible control schemes, derivative controllers are more effective when the underlying states change rapidly, integral controllers play more significant roles when lower-dimensional embedding structures can be constructed in accumulated states. Such flexibility in control design broadens the applicability of the control framework across a variety of trustworthy ML applications.

This work paves the way for the development of robust large language models. Presently, many large language models face challenges related to trustworthiness, including biases against minority groups in natural language generation tasks. In principle, by constructing state embedding manifolds that capture desired model behaviors, the PID control framework can be employed to adjust any unwanted behaviors in the model. This idea is similar to prompt engineering techniques used to modify input strings for achieving specific outcomes from models. These avenues will be explored further in future research.

**How this complements the research on adversarial attacks?** The PID control framework leads to a new method to generate adversarial attacks. In the current work, the aim is to improve model robustness by minimizing the objective function defined in Equation 2. On the contrary, maximizing this loss w.r.t. some input perturbation is equivalent to generating adversarial examples. This can be an optimal control-based adversarial attack algorithm.

**Can the same method be used for vision problems?** This method is applicable to computer vision problems. Typically, in deep convolutional neural networks, both the input and hidden states lie in extremely high-dimensional spaces, where the embedding manifolds for these states tend to exist. This assumption aligns with the "manifold hypothesis," which is based on the characteristics of real-world image data, and is further supported by empirical evidence as demonstrated in the studies by Chen et al. (2020) and Chen et al. (2022). Once the embedding manifolds for both input and hidden states are constructed, it becomes possible to formulate the optimal control objective function as outlined in Equations 2 and 3. By aiming to minimize this objective function, there is a potential to significantly improve the robustness of the model.

**How does this complement the existing trustworthy ML literature?** The current body of research on trustworthy machine learning predominantly emphasizes adversarial training, which leads to two significant challenges. Firstly, the process of modifying model parameters with adversarial examples demands extensive computational resources. In the context of natural language processing tasks, identifying an adversarial example typically entails solving a combinatorial optimization problem, which suffers from an exponential growth in the number of feasible solutions as the size of the problem increases. Secondly, adversarial training's efficacy diminishes in the face of unexpected adversarial attacks. This shortcoming is especially evident in the real-world application of large language models, where predicting potential adversarial attacks beforehand is unfeasible. The suggested PID control framework is designed to overcome these challenges by offering two key advantages: 1) It does not significantly increase the inference time when compared to the base model, and 2) It leverages the embedding structure of unperturbed states, making it robust against various unforeseen adversarial attacks.

**Extension to other trustworthy issues.** The proposed PID framework focuses on improving the robustness of pre-trained models through a set of linear embedding subspaces. These subspaces effectively encapsulate the embedding structure of the underlying states. This framework may be broadened to tackle various trustworthy concerns in machine learning, including issues related to fairness (Chen et al., 2023). To achieve this, it is necessary to develop embedding subspaces that are capable of capturing embedding structures that are invariant against demographic information. Our next objective is to demonstrate how the PID control framework can be adapted to manage and resolve fairness-related challenges.

**Analytic solution on the nonlinear dynamics.** In our approach, we determine the optimal control solution using an analytical method as outlined in Proposition 3. This analytical method is based on the assumption that the layer transformation linearization in the pre-trained model is orthogonal. Consequently, this leads to a time-varying control regularization across various layers, which is independent of the pre-trained model. This method has yielded satisfactory empirical outcomes. Nonetheless, there is a need for a more precise analytical solution that accounts for the intrinsic aspects of the underlying model. To achieve a more refined analytical solution for the optimal control issue, our next step involves linearizing the non-linear layer transformations in the pre-trained model and subsequently utilizing the Riccati equation to generate the optimal control solution.

## 6 Conclusion

Our study has introduced a novel PID control framework to improve neural network robustness against (unforeseen) input perturbations, outperforming traditional adversarial training methods. This approach maintains computational efficiency, enhances robustness in large language models, and allows for rapid online inference. Our comprehensive error analysis has confirmed the framework's effectiveness in simulated environments, contributing significantly to neural network security and robustness, and paving the way for more reliable NLP models in critical applications.

## 7 Acknowledgements

Z. Chen and Z. Zhang are supported by the NSF grant #2107321 under the CCF division. Q. Li is supported by the National Research Foundation, Singapore, under the NRF fellowship (project No. NRF-NRFF13-2021-0005).

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

## 8 Appendix A

In this section, we elaborate on the derivation of the analytic solution as presented in Propositions 1 and 3. Let $\boldsymbol{\theta}_t$ represent the $t^{\text{th}}$ linear transformation, and $\pi_t : \mathbb{R}^d \to \mathbb{R}^d$ denote the PID controller. The controlled dynamical system can be expressed as:

$$\mathbf{x}_{t+1} = \boldsymbol{\theta}_t(\mathbf{x}_t + \pi_t(\mathbf{x}_t)),$$

where the control action is added to the current state. Recall the running loss defined in equation 3,

$$\mathcal{L}(\{\mathbf{x}_s\}_{s=0}^t, \pi_t, (f_t^P, f_t^I, f_t^D))$$

$$:= \frac{1}{2}\|f_t^P(\mathbf{x}_t + \pi_t(\mathbf{x}_t))\|_2^2 + \frac{1}{2}\|f_t^I(\mathbf{x}_t + \pi_t(\mathbf{x}_t) + \sum_{s=0}^{t-1}\mathbf{x}_s)\|_2^2 + \frac{1}{2}\|f_t^D(\mathbf{x}_t + \pi_t(\mathbf{x}_t) - \mathbf{x}_{t-1})\|_2^2 + \frac{c_t}{2}\|\pi_t(\mathbf{x}_t)\|_2^2,$$

we consider the surjective mappings $f_t^P$, $f_t^I$, and $f_t^D$ as orthogonal projections. Let $\mathbf{Q}_t^P$, $\mathbf{Q}_t^I$, and $\mathbf{Q}_t^D$ be the orthogonal projections associated with $f_t^P$, $f_t^I$, and $f_t^D$, respectively, assuming a uniformly bounded state space with $\max_{\mathbf{x}\in\mathcal{X}}\|\mathbf{x}\|_2^2 \leq B$, the running loss can be bounded as follows,

$$\mathcal{L}(\{\mathbf{x}_s\}_{s=0}^t, \pi_t, (\mathbf{Q}_t^P, \mathbf{Q}_t^I, \mathbf{Q}_t^D))$$

$$= \frac{1}{2}\|\mathbf{Q}_t^P(\mathbf{x}_t + \pi_t(\mathbf{x}_t))\|_2^2 + \frac{1}{2}\|\mathbf{Q}_t^I(\mathbf{x}_t + \pi_t(\mathbf{x}_t) + \sum_{s=0}^{t-1}\mathbf{x}_s)\|_2^2 + \frac{1}{2}\|\mathbf{Q}_t^D(\mathbf{x}_t + \pi_t(\mathbf{x}_t) - \mathbf{x}_{t-1})\|_2^2 + \frac{c_t}{2}\|\pi_t(\mathbf{x}_t)\|_2^2,$$

$$\leq \frac{1}{2}\|\mathbf{Q}_t^P(\mathbf{x}_t + \pi_t(\mathbf{x}_t))\|_2^2 + \frac{1}{2}\|\mathbf{Q}_t^I(\mathbf{x}_t + \pi_t(\mathbf{x}_t))\|_2^2 + \frac{1}{2}\|\mathbf{Q}_t^I(\sum_{s=0}^{t-1}\mathbf{x}_s)\|_2^2 + \frac{1}{2}\|\mathbf{Q}_t^D(\mathbf{x}_t + \pi_t(\mathbf{x}_t))\|_2^2 + \frac{1}{2}\|\mathbf{Q}_t^D\mathbf{x}_{t-1}\|_2^2$$

$$\quad + \frac{c_t}{2}\|\pi_t(\mathbf{x}_t)\|_2^2,$$

$$\leq \frac{1}{2}\|\mathbf{Q}_t^P(\mathbf{x}_t + \pi_t(\mathbf{x}_t))\|_2^2 + \frac{1}{2}\|\mathbf{Q}_t^I(\mathbf{x}_t + \pi_t(\mathbf{x}_t))\|_2^2 + \frac{1}{2}\|\mathbf{Q}_t^D(\mathbf{x}_t + \pi_t(\mathbf{x}_t))\|_2^2 + \frac{1}{2}\|\sum_{s=0}^{t-1}\mathbf{x}_s\|_2^2 + \frac{1}{2}\|\mathbf{x}_{t-1}\|_2^2$$

$$\quad + \frac{c_t}{2}\|\pi_t(\mathbf{x}_t)\|_2^2,$$

$$\leq \frac{1}{2}\|\mathbf{Q}_t^P(\mathbf{x}_t + \pi_t(\mathbf{x}_t))\|_2^2 + \frac{1}{2}\|\mathbf{Q}_t^I(\mathbf{x}_t + \pi_t(\mathbf{x}_t))\|_2^2 + \frac{1}{2}\|\mathbf{Q}_t^D(\mathbf{x}_t + \pi_t(\mathbf{x}_t))\|_2^2 + \frac{c_t}{2}\|\pi_t(\mathbf{x}_t)\|_2^2 + \frac{TB}{2}, \tag{10}$$

where $T$ represents the maximum number of layers of the neural network, and $B$ is the uniform upper bound for the state space.

Let $\mathbf{Q}_t = \mathbf{Q}_t^P + \mathbf{Q}_t^I + \mathbf{Q}_t^D$, the following Lemma derives the analytic solution for the PID control $\pi_t(\mathbf{x_t})$.

**Proposition 5.** *Consider the following objective function,*

$$\min_{\overline{\pi}} \mathbb{E}_{(\mathbf{x}_0, y)\sim\mathcal{D}}[J(\mathbf{x}_0, y, \overline{\pi})] := \min_{\overline{\pi}} \mathbb{E}_{(\mathbf{x}_0, y)\sim\mathcal{D}}\left[\Phi(\mathbf{x}_T, y) + \sum_{t=0}^{T-1}\mathcal{L}(\{\mathbf{x}_s\}_{s=0}^t, \pi_t, (\mathbf{Q}_t^P, \mathbf{Q}_t^I, \mathbf{Q}_t^D))\right],$$

$$\text{s.t. } \mathbf{x}_{t+1} = \boldsymbol{\theta}_t(\mathbf{x}_t + \pi_t(\mathbf{x}_t)). \tag{7}$$

*the optimal value function, parametrized as $V(\mathbf{x}_t) = \mathbf{x}_t^\top \mathbf{P}_t \mathbf{x}_t$, satisfies the Riccati equation:*

$$\mathbf{P}_t = \frac{1}{2}\mathbf{Q}_t + \boldsymbol{\theta}_t^\top \mathbf{P}_{t+1}\boldsymbol{\theta}_t - \frac{1}{2}(\mathbf{Q}_t + 2\boldsymbol{\theta}_t^\top \mathbf{P}_{t+1}\boldsymbol{\theta}_t)^\top (\mathbf{Q}_t + 2\boldsymbol{\theta}_t^\top \mathbf{P}_{t+1}\boldsymbol{\theta}_t + c_t\mathbf{I})^{-1}(\mathbf{Q}_t + 2\boldsymbol{\theta}_t^\top \mathbf{P}_{t+1}\boldsymbol{\theta}_t). \tag{8}$$

*The optimal control solution is given by*

$$\pi_t(\mathbf{x}_t) = -(\mathbf{Q}_t + c\cdot\mathbf{I} + 2\boldsymbol{\theta}_t^\top \mathbf{P}_{t+1}\boldsymbol{\theta}_t)^{-1}(\mathbf{Q}_t + 2\boldsymbol{\theta}_t^\top \mathbf{P}_{t+1}\boldsymbol{\theta}_t)\mathbf{x}_t, \tag{9}$$

*where $\mathbf{Q}_t = \mathbf{Q}_t^P + \mathbf{Q}_t^I + \mathbf{Q}_t^D$.*

*Proof.* In the objective function defined in equation 7, the terminal loss $\Phi(\mathbf{x}_T, y)$ quantifies the discrepancy between the terminal state $\mathbf{x}_T$ and the true label $y$. However, in general machine learning applications, the true label $y$ remains unknown during online inference, leading to the terminal loss being set to zero. Recall equation 10, the running loss is defined as

$$\mathcal{L}(\{\mathbf{x}_s\}_{s=0}^t, \pi_t, (\mathbf{Q}_t^P, \mathbf{Q}_t^I, \mathbf{Q}_t^D))$$
$$:= \frac{1}{2}\|\mathbf{Q}_t^P(\mathbf{x}_t + \pi_t(\mathbf{x}_t))\|_2^2 + \frac{1}{2}\|\mathbf{Q}_t^I(\mathbf{x}_t + \pi_t(\mathbf{x}_t) + \sum_{s=0}^{t-1}\mathbf{x}_s)\|_2^2 + \frac{1}{2}\|\mathbf{Q}_t^D(\mathbf{x}_t + \pi_t(\mathbf{x}_t) - \mathbf{x}_{t-1})\|_2^2 + \frac{c_t}{2}\|\pi_t(\mathbf{x}_t)\|_2^2.$$

Consequently, the optimal value function $V(\mathbf{x}_t)$ satisfies

$$V(\mathbf{x}_t) = \min_{\pi_t} \frac{1}{2}(\mathbf{Q}_t^P\mathbf{x}_t + \mathbf{Q}_t^P\pi_t(\mathbf{x}_t))^\top(\mathbf{Q}_t^P\mathbf{x}_t + \mathbf{Q}_t^P\pi_t(\mathbf{x}_t)) + \frac{1}{2}(\mathbf{Q}_t^I\mathbf{x}_t + \mathbf{Q}_t^I\pi_t(\mathbf{x}_t))^\top(\mathbf{Q}_t^I\mathbf{x}_t + \mathbf{Q}_t^I\pi_t(\mathbf{x}_t))$$
$$+\frac{1}{2}(\mathbf{Q}_t^D\mathbf{x}_t + \mathbf{Q}_t^D\pi_t(\mathbf{x}_t))^\top(\mathbf{Q}_t^D\mathbf{x}_t + \mathbf{Q}_t^D\pi_t(\mathbf{x}_t)) + \frac{c}{2}\cdot\pi_t(\mathbf{x}_t)^\top\pi_t(\mathbf{x}_t) + V(\mathbf{x}_{t+1}),$$
$$\text{s.t. } \mathbf{x}_{t+1} = \boldsymbol{\theta}_t(\mathbf{x}_t + \pi_t(\mathbf{x}_t)).$$

Taking the derivative of the right-hand side with respect to $\pi_t(\mathbf{x}_t)$ yields

$$\frac{dV(\mathbf{x}_t)}{d\pi_t(\mathbf{x}_t)} = \mathbf{Q}_t^P\mathbf{x}_t + \mathbf{Q}_t^P\pi_t(\mathbf{x}_t) + \mathbf{Q}_t^I\mathbf{x}_t + \mathbf{Q}_t^I\pi_t(\mathbf{x}_t) + \mathbf{Q}_t^D\mathbf{x}_t + \mathbf{Q}_t^D\pi_t(\mathbf{x}_t) + c\pi_t(\mathbf{x}_t) + \left(\frac{d\mathbf{x}_{t+1}}{d\pi_t(\mathbf{x}_t)}\right)^\top\frac{dV(\mathbf{x}_{t+1})}{d\mathbf{x}_{t+1}},$$
$$= \mathbf{Q}_t^P\mathbf{x}_t + \mathbf{Q}_t^P\pi_t(\mathbf{x}_t) + \mathbf{Q}_t^I\mathbf{x}_t + \mathbf{Q}_t^I\pi_t(\mathbf{x}_t) + \mathbf{Q}_t^D\mathbf{x}_t + \mathbf{Q}_t^D\pi_t(\mathbf{x}_t) + c\pi_t(\mathbf{x}_t) + 2\boldsymbol{\theta}_t^\top\mathbf{P}_{t+1}\mathbf{x}_{t+1},$$
$$= (\mathbf{Q}_t^P + \mathbf{Q}_t^I + \mathbf{Q}_t^D)\mathbf{x}_t + (\mathbf{Q}_t^P + \mathbf{Q}_t^I + \mathbf{Q}_t^D)\pi_t(\mathbf{x}_t) + c\pi_t(\mathbf{x}_t) + 2\boldsymbol{\theta}_t^\top\mathbf{P}_{t+1}\boldsymbol{\theta}_t\mathbf{x}_t + 2\boldsymbol{\theta}_t^\top\mathbf{P}_{t+1}\boldsymbol{\theta}_t\pi_t(\mathbf{x}_t),$$
$$= \mathbf{Q}_t\mathbf{x}_t + \mathbf{Q}_t\pi_t(\mathbf{x}_t) + c\pi_t(\mathbf{x}_t) + 2\boldsymbol{\theta}_t^\top\mathbf{P}_{t+1}\boldsymbol{\theta}_t\mathbf{x}_t + 2\boldsymbol{\theta}_t^\top\mathbf{P}_{t+1}\boldsymbol{\theta}_t\pi_t(\mathbf{x}_t),$$

where $\mathbf{Q}_t = \mathbf{Q}_t^P + \mathbf{Q}_t^I + \mathbf{Q}_t^D$.

Setting the derivative $\frac{dV(\mathbf{x}_t)}{d\pi_t(\mathbf{x}_t)}$ to $\mathbf{0}$ results in the optimal control $\pi_t^*(\mathbf{x}_t)$ (as shown in equation 9)

$$\pi_t^*(\mathbf{x}_t) = -(\mathbf{Q}_t + c\cdot\mathbf{I} + 2\boldsymbol{\theta}_t^\top\mathbf{P}_{t+1}\boldsymbol{\theta}_t)^{-1}(\mathbf{Q}_t + 2\boldsymbol{\theta}_t^\top\mathbf{P}_{t+1}\boldsymbol{\theta}_t)\mathbf{x}_t.$$

Parametrizing the value function $V(\mathbf{x}_t)$ as $\mathbf{x}_t^\top\mathbf{P}_t\mathbf{x}_t$ and considering the optimal control solution equation 9, we can convert the expression of the value function as follows,

$$\mathbf{x}_t^\top\mathbf{P}_t\mathbf{x}_t$$
$$= \min_{\pi_t} \frac{1}{2}(\mathbf{Q}_t^P\mathbf{x}_t + \mathbf{Q}_t^P\pi_t(\mathbf{x}_t))^\top(\mathbf{Q}_t^P\mathbf{x}_t + \mathbf{Q}_t^P\pi_t(\mathbf{x}_t)) + \frac{1}{2}(\mathbf{Q}_t^I\mathbf{x}_t + \mathbf{Q}_t^I\pi_t(\mathbf{x}_t))^\top(\mathbf{Q}_t^I\mathbf{x}_t + \mathbf{Q}_t^I\pi_t(\mathbf{x}_t))$$
$$+ \frac{1}{2}(\mathbf{Q}_t^D\mathbf{x}_t + \mathbf{Q}_t^D\pi_t(\mathbf{x}_t))^\top(\mathbf{Q}_t^D\mathbf{x}_t + \mathbf{Q}_t^D\pi_t(\mathbf{x}_t)) + \frac{c}{2}\cdot\pi_t(\mathbf{x}_t)^\top\pi_t(\mathbf{x}_t) + \mathbf{x}_{t+1}^\top\mathbf{P}_{t+1}\mathbf{x}_{t+1},$$
$$= \frac{1}{2}\mathbf{x}_t^\top(\mathbf{Q}_t^P + \mathbf{Q}_t^I + \mathbf{Q}_t^D + 2\boldsymbol{\theta}_t^\top\mathbf{P}_{t+1}\boldsymbol{\theta}_t)\mathbf{x}_t + \frac{1}{2}(\pi_t^*(\mathbf{x}_t))^\top(\mathbf{Q}_t^P + \mathbf{Q}_t^I + \mathbf{Q}_t^D + c\mathbf{I} + 2\boldsymbol{\theta}_t^\top\mathbf{P}_{t+1}\boldsymbol{\theta}_t)\pi_t^*(\mathbf{x}_t)$$
$$+ \mathbf{x}_t^\top(\mathbf{Q}_t^P + \mathbf{Q}_t^I + \mathbf{Q}_t^D + 2\boldsymbol{\theta}_t^\top\mathbf{P}_{t+1}\boldsymbol{\theta}_t)\pi_t^*(\mathbf{x}_t),$$
$$= \frac{1}{2}\mathbf{x}_t^\top(\mathbf{Q}_t + 2\boldsymbol{\theta}_t^\top\mathbf{P}_{t+1}\boldsymbol{\theta}_t)\mathbf{x}_t + \frac{1}{2}(\pi_t^*(\mathbf{x}_t))^\top(\mathbf{Q}_t + c\mathbf{I} + 2\boldsymbol{\theta}_t^\top\mathbf{P}_{t+1}\boldsymbol{\theta}_t)\pi_t^*(\mathbf{x}_t) + \mathbf{x}_t^\top(\mathbf{Q}_t + 2\boldsymbol{\theta}_t^\top\mathbf{P}_{t+1}\boldsymbol{\theta}_t)\pi_t^*(\mathbf{x}_t),$$

where $\pi_t^*(\mathbf{x}_t)$ is the optimal control solution leading to the minimum, $\mathbf{Q}_t = \mathbf{Q}_t^P + \mathbf{Q}_t^I + \mathbf{Q}_t^D$. For the second term in the above, recall the optimal control solution $\pi_t^*(\mathbf{x}_t)$ from equation 9,

$$\frac{1}{2}(\pi_t^*(\mathbf{x}_t))^\top(\mathbf{Q}_t + c\cdot\mathbf{I} + 2\boldsymbol{\theta}_t^\top\mathbf{P}_{t+1}\boldsymbol{\theta}_t)\pi_t^*(\mathbf{x}_t),$$
$$= -\frac{1}{2}\left((\mathbf{Q}_t + c\cdot\mathbf{I} + 2\boldsymbol{\theta}_t^\top\mathbf{P}_{t+1}\boldsymbol{\theta}_t)^{-1}(\mathbf{Q}_t + 2\boldsymbol{\theta}_t^\top\mathbf{P}_{t+1}\boldsymbol{\theta}_t)\mathbf{x}_t\right)^\top(\mathbf{Q}_t + c + 2\boldsymbol{\theta}_t^\top\mathbf{P}_{t+1}\boldsymbol{\theta}_t)\pi_t^*(\mathbf{x}_t),$$
$$= -\frac{1}{2}\mathbf{x}_t^\top(\mathbf{Q}_t + 2\boldsymbol{\theta}_t^\top\mathbf{P}_{t+1}\boldsymbol{\theta}_t)\pi_t^*(\mathbf{x}_t),$$

the above uses the fact that $(\mathbf{Q}_t + c \cdot \mathbf{I} + 2\boldsymbol{\theta}_t^\top \mathbf{P}_{t+1}\boldsymbol{\theta}_t)^{-1}$ is symmetric. Therefore,

$$
\begin{aligned}
& \mathbf{x}_t^\top \mathbf{P}_t \mathbf{x}_t \\
&= \frac{1}{2}\mathbf{x}_t^\top (\mathbf{Q}_t + 2\boldsymbol{\theta}_t^\top \mathbf{P}_{t+1}\boldsymbol{\theta}_t)\mathbf{x}_t - \frac{1}{2}\mathbf{x}_t^\top (\mathbf{Q}_t + 2\boldsymbol{\theta}_t^\top \mathbf{P}_{t+1}\boldsymbol{\theta}_t)\pi_t^*(\mathbf{x}_t) + \mathbf{x}_t^\top (\mathbf{Q}_t + 2\boldsymbol{\theta}_t^\top \mathbf{P}_{t+1}\boldsymbol{\theta}_t)\pi_t^*(\mathbf{x}_t), \\
&= \frac{1}{2}\mathbf{x}_t^\top (\mathbf{Q}_t + 2\boldsymbol{\theta}_t^\top \mathbf{P}_{t+1}\boldsymbol{\theta}_t)\mathbf{x}_t + \frac{1}{2}\mathbf{x}_t^\top (\mathbf{Q}_t^\top \mathbf{Q}_t + 2\boldsymbol{\theta}_t^\top \mathbf{P}_{t+1}\boldsymbol{\theta}_t)\pi_t^*(\mathbf{x}_t),
\end{aligned}
$$

which results in the algebraic Riccati equation

$$
\mathbf{P}_t = \frac{1}{2}\mathbf{Q}_t + \boldsymbol{\theta}_t^\top \mathbf{P}_{t+1}\boldsymbol{\theta}_t - \frac{1}{2}(\mathbf{Q}_t + 2\boldsymbol{\theta}_t^\top \mathbf{P}_{t+1}\boldsymbol{\theta}_t)^\top (\mathbf{Q}_t + 2\boldsymbol{\theta}_t^\top \mathbf{P}_{t+1}\boldsymbol{\theta}_t + c\mathbf{I})^{-1}(\mathbf{Q}_t + 2\boldsymbol{\theta}_t^\top \mathbf{P}_{t+1}\boldsymbol{\theta}_t).
$$

$\square$

In our analysis, we focus on a specific scenario where each linear transformation $\boldsymbol{\theta}_t$ is both orthogonal and full-rank. This implies that the linear transformations satisfy the condition $\boldsymbol{\theta}_t^\top \boldsymbol{\theta}_t = \boldsymbol{\theta}_t \boldsymbol{\theta}_t^\top = \mathbf{I}$ for all $t$ in the considered range.

Recall that $\mathbf{Q}_t = \mathbf{Q}_t^P + \mathbf{Q}_t^I + \mathbf{Q}_t^D$, where

$$
\mathbf{Q}_t^P = \mathbf{I} - \mathbf{V}_t^P(\mathbf{V}_t^P)^\top, \quad \mathbf{Q}_t^I = \mathbf{I} - \mathbf{V}_t^I(\mathbf{V}_t^I)^\top, \quad \mathbf{Q}_t^D = \mathbf{I} - \mathbf{V}_t^D(\mathbf{V}_t^D)^\top,
$$

are orthogonal projections corresponding to linear embedding subspaces of state, state integration, and state derivative. For simplicity, we assume that the basis $\mathbf{V}_t^P$, $\mathbf{V}_t^I$, and $\mathbf{V}_t^D$ are mutually orthogonal to each other, meaning that

$$
(\mathbf{V}_t^P)^\top \mathbf{V}_t^I = \mathbf{0}, \quad (\mathbf{V}_t^P)^\top \mathbf{V}_t^D = \mathbf{0}, \quad (\mathbf{V}_t^I)^\top \mathbf{V}_t^D = \mathbf{0}.
$$

Based on this assumption, the combination of three orthogonal projections $\mathbf{Q}_t$ is an orthogonal projection,

$$
\begin{aligned}
\mathbf{Q}_t &= \mathbf{V}_t^P \begin{bmatrix} 0 & 0 & \cdots & 0 & 0 \\ 0 & 0 & \cdots & 0 & 0 \\ \vdots & \vdots & \ddots & 0 & 0 \\ 0 & 0 & \cdots & 1 & 0 \\ 0 & 0 & \cdots & 0 & 1 \end{bmatrix}(\mathbf{V}_t^P)^\top + \mathbf{V}_t^I \begin{bmatrix} 0 & 0 & \cdots & 0 & 0 \\ 0 & 0 & \cdots & 0 & 0 \\ \vdots & \vdots & \ddots & 0 & 0 \\ 0 & 0 & \cdots & 1 & 0 \\ 0 & 0 & \cdots & 0 & 1 \end{bmatrix}(\mathbf{V}_t^I)^\top + \mathbf{V}_t^D \begin{bmatrix} 0 & 0 & \cdots & 0 & 0 \\ 0 & 0 & \cdots & 0 & 0 \\ \vdots & \vdots & \ddots & 0 & 0 \\ 0 & 0 & \cdots & 1 & 0 \\ 0 & 0 & \cdots & 0 & 1 \end{bmatrix}(\mathbf{V}_t^D)^\top, \\
&= \mathbf{V}_t \begin{bmatrix} 0 & 0 & \cdots & 0 & 0 \\ 0 & 0 & \cdots & 0 & 0 \\ \vdots & \vdots & \ddots & 0 & 0 \\ 0 & 0 & \cdots & 1 & 0 \\ 0 & 0 & \cdots & 0 & 1 \end{bmatrix}\mathbf{V}_t^\top,
\end{aligned}
$$

where $\mathbf{V}_t$ represents the basis for the intersection of $\mathbf{V}_t^P$, $\mathbf{V}_t^I$, and $\mathbf{V}_t^D$.

**Lemma 6.** *Consider a T-layer neural network characterized by orthogonal linear transformations. The solution to the algebraic Riccati equation, as delineated in equation 8, is given by*

$$
\mathbf{P}_t = \frac{1}{2}\mathbf{V}_t \begin{bmatrix} 0 & 0 & \cdots & 0 & 0 \\ 0 & 0 & \cdots & 0 & 0 \\ \vdots & \vdots & \ddots & 0 & 0 \\ 0 & 0 & \cdots & \lambda_t & 0 \\ 0 & 0 & \cdots & 0 & \lambda_t \end{bmatrix}\mathbf{V}_t^\top, \tag{11}
$$

*where the parameter $\lambda_t$ is governed by a backward difference equation $\lambda_t = \frac{c(1+\lambda_{t+1})}{1+\lambda_{t+1}+c}$, with the initial condition specified as $\lambda_T = 0$.*

*Proof.* The proof proceeds by induction on $t$. Recall the algebraic Riccati equation 8. Given the terminal condition $\mathbf{P}_T = \mathbf{0}$, the equation for $t = T - 1$ is

$$\mathbf{P}_{T-1} = \frac{1}{2}\mathbf{Q}_{T-1} - \frac{1}{2}\mathbf{Q}_{T-1}^\top(\mathbf{Q}_{T-1} + c\mathbf{I})^{-1}\mathbf{Q}_{T-1},$$

$$= \frac{1}{2}\mathbf{V}_{T-1}\begin{bmatrix} 0 & 0 & \cdots & 0 & 0 \\ 0 & 0 & \cdots & 0 & 0 \\ \vdots & \vdots & \ddots & 0 & 0 \\ 0 & 0 & \cdots & \frac{c}{1+c} & 0 \\ 0 & 0 & \cdots & 0 & \frac{c}{1+c} \end{bmatrix}\mathbf{V}_{T-1}^\top,$$

Suppose it is true for $t + 1$, such that,

$$\mathbf{P}_{t+1} = \frac{1}{2}\mathbf{V}_{t+1}\begin{bmatrix} 0 & 0 & \cdots & 0 & 0 \\ 0 & 0 & \cdots & 0 & 0 \\ \vdots & \vdots & \ddots & 0 & 0 \\ 0 & 0 & \cdots & \lambda_{t+1} & 0 \\ 0 & 0 & \cdots & 0 & \lambda_{t+1} \end{bmatrix}\mathbf{V}_{t+1}^\top.$$

Given that $\boldsymbol{\theta}_t^\top\boldsymbol{\theta}_t = \boldsymbol{\theta}_t\boldsymbol{\theta}_t^\top = \mathbf{I}$, $\boldsymbol{\theta}_t^\top\mathbf{V}_{t+1} = \mathbf{V}_t$, in which case, $\mathbf{Q}_t$ and $\boldsymbol{\theta}_t^\top\mathbf{P}_{t+1}\boldsymbol{\theta}_t$ contain the same basis $\mathbf{V}_t$. Recall the algebraic Riccati equation 8,

$$\mathbf{P}_t = \frac{1}{2}\mathbf{Q}_t + \boldsymbol{\theta}_t^\top\mathbf{P}_{t+1}\boldsymbol{\theta}_t - \frac{1}{2}(\mathbf{Q}_t + 2\boldsymbol{\theta}_t^\top\mathbf{P}_{t+1}\boldsymbol{\theta}_t)^\top(\mathbf{Q}_t + 2\boldsymbol{\theta}_t^\top\mathbf{P}_{t+1}\boldsymbol{\theta}_t + c\mathbf{I})^{-1}(\mathbf{Q}_t + 2\boldsymbol{\theta}_t^\top\mathbf{P}_{t+1}\boldsymbol{\theta}_t),$$

$$= \frac{1}{2}\mathbf{V}_t\begin{bmatrix} 0 & \cdots & 0 \\ 0 & \cdots & 0 \\ \vdots & \ddots & 0 \\ 0 & \cdots & 1 + \lambda_{t+1} \end{bmatrix}\mathbf{V}_t^\top - \frac{1}{2}\mathbf{V}_t\begin{bmatrix} 0 & \cdots & 0 \\ 0 & \cdots & 0 \\ \vdots & \ddots & 0 \\ 0 & \cdots & (1 + \lambda_{t+1})^2(1 + \lambda_{t+1} + c)^{-1} \end{bmatrix}\mathbf{V}_t^\top,$$

$$= \frac{1}{2}\mathbf{V}_t\begin{bmatrix} 0 & 0 & \cdots & 0 \\ 0 & 0 & \cdots & 0 \\ \vdots & \vdots & \ddots & 0 \\ 0 & 0 & \cdots & \lambda_t = \frac{c(1+\lambda_{t+1})}{1+\lambda_{t+1}+c} \end{bmatrix}\mathbf{V}_t^\top.$$

$\square$

Recall the optimal control solution in equation 9 and Lemma 6, we reach the following analytic formulation.

**Proposition 7.** *When the layer-wise transformations are represented as orthogonal matrices, and the basis of state embedding, state integration embedding, and state derivative embeddings are mutually orthogonal, the optimal feedback control, denoted as $\pi_t(\mathbf{x}_t)$, can be computed as follows:*

$$\pi_t(\mathbf{x}_t) = -\mathbf{V}_t\begin{bmatrix} 0 & 0 & \cdots & 0 & 0 \\ 0 & 0 & \cdots & 0 & 0 \\ \vdots & \vdots & \ddots & 0 & 0 \\ 0 & 0 & \cdots & 1 - \frac{c}{1+\lambda_{t+1}+c} & 0 \\ 0 & 0 & \cdots & 0 & 1 - \frac{c}{1+\lambda_{t+1}+c} \end{bmatrix}\mathbf{V}_t^\top\mathbf{x}_t,$$

*where the time-varying parameter $\lambda_t$ is governed by a backward difference equation $\lambda_t = \frac{c(1+\lambda_{t+1})}{1+\lambda_{t+1}+c}$, with the terminal condition specified as $\lambda_T = 0$.*

## 9   Appendix B

Recall the optimal control formulation in Proposition 3, we define a control gain matrix $\mathbf{K}_t$

$$\mathbf{K}_t = -\mathbf{V}_t \begin{bmatrix} 0 & 0 & \cdots & & 0 \\ 0 & 0 & \cdots & & 0 \\ \vdots & \vdots & \ddots & & 0 \\ 0 & 0 & \cdots & & 1 - \frac{c}{1+\lambda_{t+1}+c} \end{bmatrix} \mathbf{V}_t^\top.$$

Let $\boldsymbol{\theta}_t$ represent the $t^{\text{th}}$ linear transformation, and $\pi : \mathbb{R}^d \to \mathbb{R}^d$ be the closed-loop controller. We denote the unperturbed state at time $t$ as $\mathbf{x}_t$, and the controlled state with perturbation $\mathbf{z}$ applied at the initial condition as $\overline{\mathbf{x}}_t$,

$$\overline{\mathbf{x}}_{t+1} = \boldsymbol{\theta}_t(\overline{\mathbf{x}}_t + \pi_t(\overline{\mathbf{x}}_t)), \quad \overline{\mathbf{x}}_0 = \mathbf{x}_t + \mathbf{z}.$$

The difference between the controlled system applied with perturbation at the initial condition and the uncontrolled system without perturbation is shown

$$\begin{aligned}
\overline{\mathbf{x}}_{t+1} - \mathbf{x}_{t+1} &= \boldsymbol{\theta}_t(\overline{\mathbf{x}}_t + \pi_t(\overline{\mathbf{x}}_t) - \mathbf{x}_t), \\
&= \boldsymbol{\theta}_t(\overline{\mathbf{x}}_{\epsilon,t} - \mathbf{K}_t \overline{\mathbf{x}}_{\epsilon,t} - \mathbf{x}_t), \\
&= \boldsymbol{\theta}_t(\mathbf{I} - \mathbf{K}_t)\overline{\mathbf{x}}_t - \boldsymbol{\theta}_t \mathbf{x}_t + \boldsymbol{\theta}_t \mathbf{K}_t \mathbf{x}_t, \\
&= \boldsymbol{\theta}_t(\mathbf{I} - \mathbf{K}_t)(\overline{\mathbf{x}}_t - \mathbf{x}_t),
\end{aligned} \tag{12}$$

where $\boldsymbol{\theta}_t \mathbf{K}_t \mathbf{x}_t = \mathbf{0}$ since $\mathbf{x}_t$ is in the null space of the control gain matrix $\mathbf{K}_t$.

**Lemma 8.** *For $t \geq 0$, we have*

$$\mathbf{I} - \mathbf{K}_t = \alpha_t \cdot \mathbf{I} + (1 - \alpha_t) \cdot \mathbf{P}_t,$$

*where* $\mathbf{P}_t := \mathbf{V}_t(\mathbf{V}_t)^\top$, $\alpha_t = \frac{c}{1+\lambda_{t+1}+c}$.

*Proof.* Recall equation 12, $(\mathbf{I} - \mathbf{K}_t)$ can be expressed as

$$\mathbf{I} - \mathbf{K}_t = \mathbf{V}_t \begin{bmatrix} 1 & 0 & \cdots & & 0 \\ 0 & 1 & \cdots & & 0 \\ \vdots & \vdots & \ddots & & 0 \\ 0 & 0 & \cdots & & \frac{c}{1+\lambda_{t+1}+c} \end{bmatrix} \mathbf{V}_t^\top,$$

where the first $r$ diagonal elements are 1, and the last $(d-r)$ diagonal elements are $\frac{c}{1+\lambda_{t+1}+c}$. By denoting the projection of first $r$ columns as $\mathbf{V}_t^r$ and last $(d-r)$ columns as $\hat{\mathbf{V}}_t^r$, it can be further shown

$$\begin{aligned}
\mathbf{I} - \mathbf{K}_t &= \mathbf{V}_t^r(\mathbf{V}_t^r)^\top + \frac{c}{1+\lambda_{t+1}+c}\left(\hat{\mathbf{V}}_t^r(\hat{\mathbf{V}}_t^r)^\top\right), \\
&= \mathbf{P}_t + \alpha_t\left(\mathbf{I} - \mathbf{P}_t\right), \\
&= \alpha_t \cdot \mathbf{I} + (1 - \alpha_t) \cdot \mathbf{P}_t,
\end{aligned}$$

where $\alpha_t = \frac{c}{1+\lambda_{t+1}+c}$. $\qquad\square$

In the presented formulation, the input state space, denoted as $Z$, is partitioned into a direct sum comprising two orthogonal subspaces. This decomposition is expressed as $Z = Z^\| \oplus Z^\perp$, where $Z^\|$ represents the linear embedding subspace, encapsulating the input data. This is characterized by the condition $\mathbf{x}_0 \in \mathcal{Z}$ for all pairs $(\mathbf{x}, y)$ sampled from the distribution $\mathcal{D}$. Concurrently, $Z^\perp$ defines the orthogonal complement of $Z^\|$. Extending this notion, the time-dependent state space at any given timestep $t$ is represented as $Z_t = Z_t^\| \oplus Z_t^\perp$.

**Lemma 9.** *For $t \geq 0$, let $\mathbf{P}_t^s$ be defined as follows,*

$$\begin{cases} \mathbf{P}_t^0 := \mathbf{P}_t, \\ \mathbf{P}_t^{(s+1)} := \boldsymbol{\theta}_{t-s-1}^{-1} \mathbf{P}_t^s \boldsymbol{\theta}_{t-s-1}, \quad s = 0, 1, \ldots, t-1. \end{cases}$$

*Then*

1. $\mathbf{P}_t^s$ *is a projection.*

2. $\mathbf{P}_t^s$ *is a projection onto $Z_{t-s}^{\|}$, i.e. $range(\mathbf{P}_t^s) = Z_{t-s}^{\|}$.*

3. *If all $\boldsymbol{\theta}_t$ are orthogonal, then $\mathbf{P}_t^t = \mathbf{P}_0$, $\forall t$, where $\mathbf{P}_0$ is the orthogonal projection onto $Z_0^{\|}$.*

*Proof.* 1. We prove it by induction on $s$ for each $t$. For $s = 0$, $\mathbf{P}_t^0 = \mathbf{P}_t$, which is a projection by its definition. Suppose it is true for $s$ such that $\mathbf{P}_t^s = \mathbf{P}_t^s \mathbf{P}_t^s$ ($\mathbf{P}$ is a projection if $\mathbf{P} = \mathbf{P}^2$), then for $(s+1)$,

$$\begin{aligned} (\mathbf{P}_t^{s+1})^2 &= \left(\boldsymbol{\theta}_{t-s-1}^{-1} \mathbf{P}_t^s \boldsymbol{\theta}_{t-s-1}\right)^2, \\ &= \boldsymbol{\theta}_{t-s-1}^{-1} \left(\mathbf{P}_t^s\right)^2 \boldsymbol{\theta}_{t-s-1}, \\ &= \boldsymbol{\theta}_{t-s-1}^{-1} \mathbf{P}_t^s \boldsymbol{\theta}_{t-s-1}, \\ &= \mathbf{P}_t^{s+1}. \end{aligned}$$

2. We prove it by induction on $s$ for each $t$. For $s = 0$, $\mathbf{P}_t^0 = \mathbf{P}_t$, which is the orthogonal projection onto $Z_t^{\|}$. Suppose that it is true for $s$ such that $\mathbf{P}_t^s$ is a projection onto $Z_{t-s}^{\|}$, then for $(s+1)$, $\mathbf{P}_t^{s+1} = \boldsymbol{\theta}_{t-s-1}^{-1} \mathbf{P}_t^s \boldsymbol{\theta}_{t-s-1}$, which implies

$$\begin{aligned} range(\mathbf{P}_t^{s+1}) &= range(\boldsymbol{\theta}_{t-s-1}^{-1} \mathbf{P}_t^s), \\ &= \{\boldsymbol{\theta}_{t-s-1}^{-1} \mathbf{x} : \mathbf{x} \in Z_{t-s}^{\|}\}, \\ &= Z_{t-s-1}^{\|}. \end{aligned}$$

3. If $\boldsymbol{\theta}_t$ is orthogonal,

$$\begin{aligned} \mathbf{P}_t^{s+1} &= \boldsymbol{\theta}_{t-s-1}^{-1} \mathbf{P}_t^s \boldsymbol{\theta}_{t-s-1}, \\ &= \boldsymbol{\theta}_{t-s-1}^T \mathbf{P}_t^s \boldsymbol{\theta}_{t-s-1}, \\ &= (\mathbf{P}_t^{s+1})^{\top}. \end{aligned}$$

$\mathbf{P}_t^{s+1}$ is a orthogonal projection onto range $Z_{t-s-1}^{\|}$. Therefore, $\mathbf{P}_t^T$ is a orthogonal projection onto $Z_0^{\|}$, orthogonal projection onto the same range is unique, $\mathbf{P}_t^T = \mathbf{P}_0$, $\forall t$.

$\square$

The following Lemma uses the concept of oblique projection to show a recursive relationship to project any $t^{th}$ state space of Eq. (12) back to the input data space.

**Lemma 10.** *Define for $0 \leq s \leq t$,*

$$\mathbf{G}_t^s := \alpha_t \cdot \mathbf{I} + (1 - \alpha_t) \mathbf{P}_t^s.$$

*Then, Eq. (12) can be written as*

$$\bar{\mathbf{x}}_t - \mathbf{x}_t = (\boldsymbol{\theta}_{t-1} \boldsymbol{\theta}_{t-2} \cdots \boldsymbol{\theta}_0)(\mathbf{G}_{t-1}^{t-1} \mathbf{G}_{t-2}^{t-2} \cdots \mathbf{G}_0^0)(\bar{\mathbf{x}}_0 - \mathbf{x}_0), \quad t \geq 1.$$

*Proof.* We prove it by induction on $t$. For $t = 1$, by the definition of $\mathbf{G}_t^s$ and transformation from Lemma 8,

$$
\begin{aligned}
\overline{\mathbf{x}}_1 - \mathbf{x}_1 &= \boldsymbol{\theta}_0(\mathbf{I} - \mathbf{K}_0)(\overline{\mathbf{x}}_0 - \mathbf{x}_0), && \text{Eq. (12)}, \\
&= \boldsymbol{\theta}_0(\alpha_0 \cdot \mathbf{I} + (1 - \alpha_0) \cdot \mathbf{P}_0)(\overline{\mathbf{x}}_0 - \mathbf{x}_0), \\
&= \boldsymbol{\theta}_0 \mathbf{G}_0^0(\overline{\mathbf{x}}_0 - \mathbf{x}_0).
\end{aligned}
$$

Suppose that it is true for $(\overline{\mathbf{x}}_t - \mathbf{x}_t)$, by Lemma 8, we have

$$
\begin{aligned}
\overline{\mathbf{x}}_{t+1} - \mathbf{x}_{t+1} &= \boldsymbol{\theta}_t(\mathbf{I} - \mathbf{K}_t)(\overline{\mathbf{x}}_t - \mathbf{x}_t), \\
&= \boldsymbol{\theta}_t(\alpha_t \cdot \mathbf{I} - (1 - \alpha_t) \cdot \mathbf{P}_t)(\overline{\mathbf{x}}_t - \mathbf{x}_t), && \text{Lemma 8}, \\
&= \boldsymbol{\theta}_t \mathbf{G}_t^0(\boldsymbol{\theta}_{t-1}\boldsymbol{\theta}_{t-2}\cdots\boldsymbol{\theta}_0)(\mathbf{G}_{t-1}^{t-1}\mathbf{G}_{t-2}^{t-2}\cdots\mathbf{G}_0^0)(\overline{\mathbf{x}}_0 - \mathbf{x}_0).
\end{aligned} \tag{13}
$$

Recall the definitions of $\mathbf{P}_t^{(s+1)} := \boldsymbol{\theta}_{t-s-1}^{-1}\mathbf{P}_t^s\boldsymbol{\theta}_{t-s-1}$, and $\mathbf{G}_t^s := \alpha_t \cdot \mathbf{I} + (1 - \alpha_t)\mathbf{P}_t^s$,

$$
\begin{aligned}
\mathbf{G}_t^{s+1} &= \alpha_t \cdot \mathbf{I} + (1 - \alpha_t) \cdot \mathbf{P}_t^{(s+1)}, \\
&= \alpha_t \cdot \mathbf{I} + (1 - \alpha_t) \cdot \boldsymbol{\theta}_{t-s-1}^{-1}\mathbf{P}_t^s\boldsymbol{\theta}_{t-s-1}, \\
&= \boldsymbol{\theta}_{t-s-1}^{-1}\big(\alpha_t \cdot \mathbf{I} + (1 - \alpha_t) \cdot \mathbf{P}_t^s\big)\boldsymbol{\theta}_{t-s-1}, \\
&= \boldsymbol{\theta}_{t-s-1}^{-1}\mathbf{G}_t^s\boldsymbol{\theta}_{t-s-1},
\end{aligned}
$$

which results in the equality for the oblique projections. Furthermore,

$$
\boldsymbol{\theta}_{t-s-1}\mathbf{G}_t^{(s+1)} = \mathbf{G}_t^s\boldsymbol{\theta}_{t-s-1}.
$$

Applying the above to Eq. (13) results in

$$
\begin{aligned}
\overline{\mathbf{x}}_{t+1} - \mathbf{x}_{t+1} &= \boldsymbol{\theta}_t\mathbf{G}_t^0(\boldsymbol{\theta}_{t-1}\boldsymbol{\theta}_{t-2}\cdots\boldsymbol{\theta}_0)(\mathbf{G}_{t-1}^{t-1}\mathbf{G}_{t-2}^{t-2}\cdots\mathbf{G}_0^0)(\overline{\mathbf{x}}_0 - \mathbf{x}_0), \\
&= (\boldsymbol{\theta}_t\boldsymbol{\theta}_{t-1})\mathbf{G}_t^1(\boldsymbol{\theta}_{t-2}\boldsymbol{\theta}_{t-3}\cdots\boldsymbol{\theta}_0)(\mathbf{G}_{t-1}^{t-1}\mathbf{G}_{t-2}^{t-2}\cdots\mathbf{G}_0^0)(\overline{\mathbf{x}}_0 - \mathbf{x}_0), \\
&= (\boldsymbol{\theta}_t\boldsymbol{\theta}_{t-1}\boldsymbol{\theta}_{t-2})\mathbf{G}_t^2(\boldsymbol{\theta}_{t-3}\boldsymbol{\theta}_{t-4}\cdots\boldsymbol{\theta}_0)(\mathbf{G}_{t-1}^{t-1}\mathbf{G}_{t-2}^{t-2}\cdots\mathbf{G}_0^0)(\overline{\mathbf{x}}_0 - \mathbf{x}_0), \\
&= (\boldsymbol{\theta}_t\boldsymbol{\theta}_{t-1}\cdots\boldsymbol{\theta}_0)(\mathbf{G}_t^t\mathbf{G}_{t-1}^{t-1}\cdots\mathbf{G}_0^0)(\overline{\mathbf{x}}_0 - \mathbf{x}_0).
\end{aligned}
$$

$\square$

**Lemma 11.** *Let*

$$
\mathbf{F}_t := \mathbf{G}_{t-1}^{(t-1)}\mathbf{G}_{t-2}^{(t-2)}\cdots\mathbf{G}_0^0, \quad t \geq 1.
$$

*Then,*

$$
\mathbf{F}_t = \prod_{s=0}^{t-1}\alpha_s \cdot \mathbf{I} + (1 - \prod_{s=0}^{t-1}\alpha_s) \cdot \mathbf{P}_0.
$$

*Proof.* We prove it by induction on $t$. Recall the definition of $\mathbf{G}_t^s := \alpha_t \cdot \mathbf{I} + (1 - \alpha_t) \cdot \mathbf{P}_t^s$. When $t = 1$,

$$
\mathbf{F}_1 = \mathbf{G}_0^0 = \alpha_0 \cdot \mathbf{I} + (1 - \alpha_0) \cdot \mathbf{P}_0.
$$

Suppose that it is true for $t$ such that

$$
\mathbf{F}_t = \prod_{s=0}^{t-1}\alpha_s \cdot \mathbf{I} + (1 - \prod_{s=0}^{t-1}\alpha_s) \cdot \mathbf{P}_0,
$$

for $(t+1)$,

$$
\begin{aligned}
\mathbf{F}_{t+1} &= \mathbf{G}_t^t \mathbf{F}_t, \\
&= (\alpha_t \cdot \mathbf{I} + (1-\alpha_t) \cdot \mathbf{P}_t^t)\mathbf{F}_t, \\
&= (\alpha_t \cdot \mathbf{I} + (1-\alpha_t) \cdot \mathbf{P}_t^t)(\prod_{s=0}^{t-1} \alpha_s \cdot \mathbf{I} + (1 - \prod_{s=0}^{t-1} \alpha_s) \cdot \mathbf{P}_0), \\
&= \prod_{s=0}^{t} \alpha_s \cdot \mathbf{I} + \alpha_t(1 - \prod_{s=0}^{t-1} \alpha_s) \cdot \mathbf{P}_0 + (1-\alpha_t)\prod_{s=0}^{t-1} \alpha_s \cdot \mathbf{P}_t^t + (1-\alpha_t)(1 - \prod_{s=1}^{t-1} \alpha_s) \cdot \mathbf{P}_t^t \mathbf{P}_0.
\end{aligned}
$$

Recall Lemma 9, if all $\boldsymbol{\theta}_t$ is orthogonal, then $\mathbf{P}_t^t = \mathbf{P}_0$, and $\mathbf{P}_t^t \mathbf{P}_0 = \mathbf{P}_0$. Hence,

$$
\begin{aligned}
\mathbf{F}_{t+1} &= \prod_{s=0}^{t} \alpha_s \cdot \mathbf{I} + \alpha_t(1 - \prod_{s=0}^{t-1} \alpha_s) \cdot \mathbf{P}_0 + (1-\alpha_t)\prod_{s=0}^{t-1} \alpha_s \cdot \mathbf{P}_0 + (1-\alpha_t)(1 - \prod_{s=1}^{t-1} \alpha_s) \cdot \mathbf{P}_0, \\
&= \prod_{s=0}^{t} \alpha_s \cdot \mathbf{I} + \left(\alpha_t - \prod_{s=0}^{t} \alpha_s + \prod_{s=0}^{t-1} \alpha_s - \prod_{s=0}^{t} \alpha_s + 1 - \alpha_t - \prod_{s=0}^{t-1} \alpha_s + \prod_{s=0}^{t} \alpha_s \right) \cdot \mathbf{P}_0, \\
&= \prod_{s=0}^{t} \alpha_s \cdot \mathbf{I} + \left(1 - \prod_{s=0}^{t} \alpha_s \right) \cdot \mathbf{P}_0.
\end{aligned}
$$

$\square$

**Theorem 4.** *For any time step $t \geq 1$, assuming that each $\boldsymbol{\theta}_t$ is an orthogonal matrix, we have the following error computation:*

$$
\|\overline{\mathbf{x}}_t - \mathbf{x}_t\|_2^2 = \prod_{s=0}^{t-1} \alpha_s^2 \cdot \|\mathbf{z}^\perp\|_2^2 + \|\mathbf{z}^\|\|_2^2,
$$

*where $\alpha_t$ is a time-varying parameter defined in relation to the control regularization $c$, and $\lambda_t$ are auxiliary variables, as follows:*

$$
\alpha_t = \frac{c}{1 + \lambda_{t+1} + c}, \quad \lambda_T = 0, \quad \lambda_{T-1} = \frac{c}{1+c}, \quad \lambda_t = \frac{c(1 + \lambda_{t+1})}{1 + c + \lambda_{t+1}}.
$$

*Proof.* The input perturbation $\mathbf{z} = \overline{\mathbf{x}}_0 - \mathbf{x}_0$ can be decomposed as $\mathbf{z} = \mathbf{z}^\| + \mathbf{z}^\perp$, where $\mathbf{z}^\| \in Z_0^\|$ and $\mathbf{z}^\perp \in Z_0^\perp$, and $\mathbf{z}^\|$ and $\mathbf{z}^\perp$ are vectors such that

- $\mathbf{z}^\| \cdot \mathbf{z}^\perp = 0$ almost surely.

- $\mathbf{z}^\|, \mathbf{z}^\perp$ have uncorrelated components.

- $\mathbf{z}^\| \in Z^\|$, and $\mathbf{z}^\perp \in \mathcal{Z}^\perp$.

Since the layer transformations $\boldsymbol{\theta}_t$ are orthogonal matrices for all $t$, recall the dynamical system Eq. (12) and Lemma 10,

$$
\begin{aligned}
\|\overline{\mathbf{x}}_t - \mathbf{x}_t\|_2^2 &= \|\boldsymbol{\theta}_t(\mathbf{I} - \mathbf{K}_t)\boldsymbol{\theta}_{t-1}(\mathbf{I} - \mathbf{K}_{t-1}) \cdots \boldsymbol{\theta}_0(\mathbf{I} - \mathbf{K}_0)\mathbf{z}\|_2^2, \\
&= \|(\boldsymbol{\theta}_{t-1}\boldsymbol{\theta}_{t-2} \cdots \boldsymbol{\theta}_0)(\mathbf{G}_{t-1}^{t-1} \cdots \mathbf{G}_0^0)\mathbf{z}\|_2^2, \\
&= \|(\mathbf{G}_{t-1}^{t-1} \cdots \mathbf{G}_0^0)\mathbf{z}\|_2^2,
\end{aligned}
\tag{14}
$$

For the term $\|(\mathbf{G}_{t-1}^{t-1}\cdots\mathbf{G}_0^0)\mathbf{z}\|_2^2$, recall Lemma 11,

$$
\begin{aligned}
&\|(\mathbf{G}_{t-1}^{t-1}\cdots\mathbf{G}_0^0)\mathbf{z}\|_2^2 \\
&= \|\bigg(\prod_{s=0}^{t-1}\alpha_s\cdot\mathbf{I}+(1-\prod_{s=0}^{t-1}\alpha_s)\mathbf{P}_0\bigg)\mathbf{z}\|_2^2, \\
&= \|\prod_{s=0}^{t-1}\alpha_s\cdot\mathbf{z}+(1-\prod_{s=0}^{t-1}\alpha_s)\cdot\mathbf{z}^\|\|_2^2, \\
&= (\prod_{s=0}^{t-1}\alpha_s)^2\cdot\|\mathbf{z}\|_2^2+(1-\prod_{s=0}^{t-1}\alpha_s)^2\cdot\|\mathbf{z}^\|\|_2^2+2(\prod_{s=0}^{t-1}\alpha_s)(1-\prod_{s=0}^{t-1}\alpha_s)(\mathbf{z})^\top\mathbf{z}^\|, \\
&= (\prod_{s=0}^{t-1}\alpha_s)^2\cdot(\|\mathbf{z}^\|\|_2^2+\|\mathbf{z}^\perp\|_2^2)+(1-\prod_{s=0}^{t-1}\alpha_0)^2\cdot\|\mathbf{z}^\|\|_2^2+2(\prod_{s=0}^{t-1}\alpha_s)(1-\prod_{s=0}^{t-1}\alpha_s)(\mathbf{z}^\|+\mathbf{z}^\perp)^\top\mathbf{z}^\| \\
&= \prod_{s=0}^{t-1}\alpha_s^2\cdot\|\mathbf{z}^\perp\|_2^2+\bigg(\prod_{s=0}^{t-1}\alpha_s^2+(1-\prod_{s=0}^{t-1}\alpha_s)^2+2(\prod_{s=0}^{t-1}\alpha_s)(1-\prod_{s=0}^{t-1}\alpha_s)\bigg)\cdot\|\mathbf{z}^\|\|_2^2, \\
&= \prod_{s=0}^{t-1}\alpha_s^2\cdot\|\mathbf{z}^\perp\|_2^2+\|\mathbf{z}^\|\|_2^2.
\end{aligned}
$$

Recall the error computation in Eq. (14),

$$
\|\overline{\mathbf{x}}_{\epsilon,t}-\mathbf{x}_t\|_2^2=\prod_{s=0}^{t-1}\alpha_s^2\cdot\|\mathbf{z}^\perp\|_2^2+\|\mathbf{z}^\|\|_2^2.
$$

$\square$

## 10 Appendix C

The tables referenced as 8 and 9 provide comprehensive numerical data for the SNLI and MNLI datasets, respectively. The results in Table 8 reveal that the PD control mechanism significantly improves the robustness of both the standard and robustly trained baseline models against every type of adversarial attack. Specifically, the application of PD control to the standardly trained Distilbert model boosts its accuracy by 15% and 16% in facing TextBugger and TextFoller attacks, respectively, while incurring a minimal 1% accuracy reduction on the original, unaltered dataset. Moreover, the robustly trained Distilbert model, which includes adversarial training (AT), benefits from the addition of PD control by showing a 12% accuracy increase when confronted with both TextBugger and TextFoller attacks. The comparative analysis of baseline models and the proposed control framework for the MNLI dataset, as detailed in Table 9, demonstrates that implementing the PD controller enhances the standard trained Distilbert's resistance to perturbations by an average of nearly 10%. However, this improvement is somewhat diminished in models trained for robustness, with an average enhancement of 5% against all types of perturbations.

## 11 Appendix D

**A optimal control framework for robust deep neural networks.** We start with a description of the dynamical system approach to machine learning. In the dynamical system framework, we consider the input $\mathbf{x}_0\in\mathcal{X}$ as the initial condition of a system of difference equations,

$$
\mathbf{x}_{t+1}=F_t(\mathbf{x}_t+\pi_t(\mathbf{x}_t),\boldsymbol{\theta}_t),
$$

where $F_t$ represents a time-varying difference equation, $\boldsymbol{\theta}_t$ are model parameters of $F_t$, $\pi_t:\mathbb{R}^d\to\mathbb{R}^d$ is a feedback controller that maps the current state $\mathbf{x}_t$ to a control action. The goal of optimal control is to

Table 8: Measurement on SNLI dataset: baseline model / controlled model

| | None | A2T | PSO | TextBugger | TextFooler |
|---|---|---|---|---|---|
| Standard models | | | | | |
| Distilbert | 87.24 / 86.05 | 53.89 / 62.31 | 49.84 / 54.96 | 24.73 / 40.26 | 24.69 / 41.73 |
| RoBERTaBase | 90.87 / 90.64 | 58.36 / 64.11 | 51.44 / 54.40 | 35.90 / 43.20 | 27.03 / 37.35 |
| BERT-large | 90.36 / 89.75 | 74.18 / 75.54 | 66.84 / 67.55 | 64.13 / 64.41 | 56.37 / 58.27 |
| RoBERTaLarge | 92.39 / 92.05 | 59.40 / 64.95 | 52.15 / 56.70 | 33.72 / 42.43 | 26.43 / 37.29 |
| Robust models (trained with AT) | | | | | |
| | None | A2T | PSO | TextBugger | TextFooler |
| Distilbert | 86.74 / 85.81 | 71.78 / 71.81 | 52.85 / 57.87 | 29.63 / 41.64 | 31.59 / 43.81 |
| RoBERTaBase | 90.65 / 89.87 | 76.28 / 77.08 | 53.85 / 56.45 | 35.43 / 43.35 | 29.64 / 39.39 |
| BERT-large | 90.29 / 90.33 | 86.02 / 85.76 | 69.23 / 70.38 | 69.17 / 69.55 | 63.78 / 65.27 |
| RoBERTaLarge | 92.10 / 91.62 | 81.11 / 81.62 | 55.28 / 59.71 | 34.15 / 44.74 | 28.74 / 42.44 |
| Robust models (trained with FreeLB) | | | | | |
| | None | A2T | PSO | TextBugger | TextFooler |
| Distilbert | 85.68 / 84.50 | 57.75 / 62.95 | 52.53 / 56.86 | 26.68 / 37.80 | 25.47 / 39.64 |
| RoBERTaBase | 91.31 / 90.67 | 64.23 / 68.85 | 52.22 / 55.24 | 34.08 / 42.75 | 24.80 / 36.81 |
| BERT-large | 90.81 / 90.72 | 77.64 / 78.21 | 64.72 / 65.56 | 58.21 / 59.29 | 53.31 / 56.26 |
| RoBERTaLarge | 92.37 / 92.26 | 67.53 / 71.30 | 53.37 / 57.20 | 34.64 / 44.42 | 27.55 / 38.59 |

Table 9: Measurement on MNLI dataset: baseline model / controlled model

| | None | A2T | PSO | TextBugger | TextFooler |
|---|---|---|---|---|---|
| Standard models | | | | | |
| Distilbert | 79.39 / 76.98 | 59.43 / 64.61 | 51.81 / 59.49 | 36.02 / 47.34 | 38.78 / 50.62 |
| RoBERTaBase | 86.66 / 85.84 | 59.60 / 63.39 | 49.77 / 53.05 | 34.76 / 40.68 | 31.43 / 40.22 |
| BERT-large | 84.92 / 84.79 | 77.38 / 77.96 | 69.71 / 70.41 | 65.11 / 65.54 | 64.80 / 65.91 |
| RoBERTaLarge | 89.71 / 89.40 | 62.85 / 67.93 | 51.19 / 56.77 | 37.18 / 45.11 | 32.81 / 43.27 |
| Robust models (trained with AT) | | | | | |
| | None | A2T | PSO | TextBugger | TextFooler |
| Distilbert | 79.70 / 76.50 | 66.52 / 67.71 | 57.34 / 62.90 | 40.22 / 50.00 | 45.62 / 54.93 |
| RoBERTaBase | 86.55 / 85.54 | 64.52 / 66.70 | 53.41 / 56.78 | 35.61 / 40.88 | 34.27 / 43.41 |
| BERT-large | 84.90 / 85.02 | 81.37 / 81.69 | 73.05 / 74.05 | 68.27 / 68.91 | 71.69 / 72.80 |
| RoBERTaLarge | 90.10 / 89.51 | 76.94 / 78.36 | 59.34 / 64.44 | 41.21 / 48.34 | 40.69 / 49.33 |
| Robust models (trained with FreeLB) | | | | | |
| | None | A2T | PSO | TextBugger | TextFooler |
| Distilbert | 78.76 / 75.33 | 64.10 / 66.25 | 58.03 / 62.94 | 38.87 / 49.50 | 43.58 / 52.88 |
| RoBERTaBase | 86.10 / 85.59 | 61.60 / 65.31 | 51.69 / 54.93 | 36.06 / 42.42 | 33.27 / 42.21 |
| BERT-large | 85.32 / 85.62 | 79.34 / 79.64 | 72.25 / 72.68 | 65.90 / 66.58 | 67.44 / 68.49 |
| RoBERTaLarge | 90.18 / 89.81 | 67.28 / 71.61 | 53.27 / 58.04 | 36.40 / 44.91 | 32.83 / 43.84 |

design these feedback controllers $\{\pi_t\}_{t=0}^{T-1}$ such that some objectives are satisfied. This can be represented as the following objective function,

$$\min_{\overline{\pi}} \mathbb{E}_{(\mathbf{x}_0, y) \sim \mathcal{D}} \left[ J(\mathbf{x}_0, \mathbf{y}, \overline{\pi}) \right] := \min_{\overline{\pi}} \mathbb{E}_{(\mathbf{x}_0, y) \sim \mathcal{D}} \left[ \Phi(\mathbf{x}_T, y) + \sum_{t=0}^{T-1} \mathcal{L}(\{\mathbf{x}_s\}_{s=0}^{t}, \pi_t, f_t) \right],$$

$$\text{s.t. } \mathbf{x}_{t+1} = F_t(\mathbf{x}_t + \pi_t(\mathbf{x}_t), \boldsymbol{\theta}_t),$$

where $\Phi(\cdot)$ and $\mathcal{L}(\cdot)$ represent terminal and running losses, respectively. In general, objectives in the real world can be structured as terminal and running loss functions. Therefore, optimizing such an objective function aligns with achieving a real-world goal. Take the development of autonomous vehicles as an example. This involves guiding the vehicle to a destination along a specific route. This challenge can be approached as an optimal control problem, where reaching the destination is expressed as a terminal loss function, and the deviation from the planned route is captured through a running loss function.

The essential task of supervised learning is to approximate some function

$$F : \mathcal{X} \to \mathcal{Y}, \quad \text{where} \quad F = F_{T-1} \circ F_{T-2} \circ \cdots \circ F_1 \circ F_0,$$

which maps inputs in $\mathcal{X} \in \mathbb{R}^d$ (e.g. images, natural language sequences) to labels in $\mathcal{Y}$ (categories, numerical predictions). The objective of developing robust deep neural networks can be formulated within an optimal control framework. Here, the aim is to minimize the discrepancy between the model's predictions and the actual labels through a terminal loss function, $\Phi(\mathbf{x}_T, y)$, by implementing feedback controllers. However, this ideal scenario is challenged by the practical limitation that true labels are unavailable during the model's inference phase. As a consequence, our focus shifts towards developing robust model predictions through the development of running loss functions. In this work, we introduce a running loss function that assesses the state of control at each timestep $t$,

$$\mathcal{L}(\{\mathbf{x}_s\}_{s=0}^t, \pi_t, (f_t^P, f_t^I, f_t^D)) := \frac{1}{2}\|f_t^P(\mathbf{x}_t + \pi_t(\mathbf{x}_t))\|_2^2 + \frac{1}{2}\|f_t^I(\mathbf{x}_t + \pi_t(\mathbf{x}_t) + \sum_{s=0}^{t-1}\mathbf{x}_s)\|_2^2$$
$$+ \frac{1}{2}\|f_t^D(\mathbf{x}_t + \pi_t(\mathbf{x}_t) - \mathbf{x}_{t-1})\|_2^2 + \frac{c_t}{2}\|\pi_t(\mathbf{x}_t)\|_2^2,$$

This running loss function calculates a loss by measuring the difference between the controlled state and certain embedding manifolds. These manifolds capture the structural, integrative, and derivative aspects of state embeddings. Ideally, the states from unperturbed input samples should align with these embedding manifolds. Thus, when perturbations are introduced to the input, this running loss function assesses the quality of the state. Minimizing this running loss helps in adjusting the states to correct the effects of such perturbations. By defining both terminal and running loss functions, solving this optimal control problem is equivalent to generating feedback controllers $\{\pi_t\}_{t=0}^{T-1}$, such that the controlled state trajectory of perturbed input performs similarly to the unperturbed counterpart.

