# OpenReview forum: "PID Control-Based Self-Healing to Improve the Robustness of Large Language Models"
_TMLR — Accepted by TMLR_

### Review · Reviewer_Tx49 · 2024-02-07

**Summary Of Contributions:**

The authors consider the problem of improving the robustness of large language models. They do this by introducing three controllers (P, I, and D) that fix the trajectory of activations of the network during inference so that it is closer to the manifold of activations observed during the training. Due to the computational complexity in this domain, they simplify the problem by assuming that the embedding manifold and layer-wise transformations are orthogonal. They show that in this case, one can efficiently find a solution. The authors then show that the method works well empirically using several BERT variants.

**Audience:**

Yes

**Claims And Evidence:**

No

**Requested Changes:**

The requested changes are closely related to the points mentioned in the weaknesses section:
* Please either run experiments on larger models or clearly state that the paper does not consider the current large (1B+ parameters) language models.
* Please discuss the differences wrt. to Chen et al's work. A discussion in the text would be good, but an empirical comparison would be ideal, in particular, to understand the trade-offs of linearization.
* Please discuss why in your opinion PD works better and why PID is used for most of the experiments instead.
* Please discuss the computation overhead of the method, preferably empirically (wall time).

**Strengths And Weaknesses:**

Strengths:
* The work aims to solve a difficult and important problem.
* The empirical results of the best-performing proposed approach are quite good.
* The theoretical derivation of the proposed solution is elegant.
* The paper is well-written and easy to read


Weaknesses:
* The title states that the work focuses on large language models, but the authors only test their methods on a few relatively small variants of BERT. These models were released at least 3 years ago and have at most a few hundred million parameters. In most cases, I would be okay with testing on smaller models, but here I'd like to argue that LLMs have undergone drastic changes in the last few years and there are hints that certain interesting behaviors only appear when we reach a few billion parameters. I realize that such experiments would be very expensive, so if these are infeasible, I would like to ask the authors to tone down the claims in the paper and clearly state that the latest large models are out of scope.
    * Additionally, I would like the authors to discuss whether the method is applicable in an autoregressive, decoder-only setting since most of the recent LLMs are decoder-only.
* The novelty is limited, as that approach operates in the framework introduced by [1, 2]. Since this is TMLR, novelty is not exactly required, but I think the differences should be more clearly highlighted in the text. Ideally, I would like to see an empirical comparison -- I understand that the P-only variant is the closest to the previous models, but the impact of linearization is not well understood. For example, I would appreciate a comparison of the PMP-based solution with the linearization-based solution on a simpler dataset, where PMP is actually applicable.
* In the end, PID seems to work worse than just PD. This seems to be inconsistent with the premise of the paper since most experiments are run on the whole PID. Can you provide a discussion on this point?
* I would appreciate a discussion of the computation overhead of the proposed method, e.g., how much slower in practice is BERT with PID compared to regular BERT?


Disclaimer: I haven't carefully checked the proofs in the Appendix.

[1] Chen, Zhuotong, Qianxiao Li, and Zheng Zhang. "Towards robust neural networks via close-loop control." arXiv preprint arXiv:2102.01862 (2021). \
[2] Chen, Zhuotong, Qianxiao Li, and Zheng Zhang. "Self-healing robust neural networks via closed-loop control." The Journal of Machine Learning Research 23.1 (2022): 14329-14382.

---

> ### Author Response · Authors · 2024-02-12
> **Reponse to the requested change of running experiments on larger models**
>
> Thank you for your insightful comments. Running large-scale models with 1B+ model parameters has been challenging with our computational resources.
> Here we present the numerical results of OPT-1.3B.
> OPT-1.3B is a decoder-based large language model that contains 1.3 billion model parameters.
> For the proposed PID control, we follow the same P-D control implementation (proportional-derivative) as done in all numerical experiments from the paper.
> The following table demonstrates that the controlled OPT-1.3B model consistently improves the robustness performance against all four types of adversarial attacks.
> Specifically,
> on the SNLI dataset,
> the average improvement is over 20% compared with the base model, with only a 2.5% accuracy drop on the unperturbed testing dataset.
> On a more challenging MNLI dataset,
> The improvement reaches 21% against the TextBugger attack, and 11% on both A2T and PSO attacks.
>
> ## SNLI Dataset
>
> |                | None             | A2T              | PSO              | TextBugger       | TextFooler       |
> |----------------|------------------|------------------|------------------|------------------|------------------|
> | **OPT**        | **91.24** / 88.69    | 49.15 / **63.28**    | 48.00 / **60.06**    | 17.57 / **41.79**    | 16.64 / **44.70**    |
>
> ## MNLI Dataset
>
> |                | None             | A2T              | PSO              | TextBugger       | TextFooler       |
> |----------------|------------------|------------------|------------------|------------------|------------------|
> | **OPT**        | **86.89** / 84.27    | 54.47 / **65.87**    | 45.14 / **59.08**    | 24.12 / **45.97**    | 21.68 / **49.13**    |

---

> ### Author Response · Authors · 2024-02-12
> **Reponse to the requested change of discussion and empirical comparison with existing works**
>
> The proposed PID control framework generalizes the existing closed-loop control approach by introducing additional derivative and integral control mechanisms.
> This work presents a fast solution to overcome the previous challenge associated with solving PMP during online inference, which is a necessary step in the existing methods.
> Detailed discussion about computational wall time is provided in response to your last request.
>
> The analytic solution relies on linearity and orthogonality assumptions.
> Although these assumptions do not generally hold, this enables the control framework to be employed in LLMs with more than 1B parameters.
> We provide an empirical comparison of the PMP and the analytic solutions in the following tables.
> These experiments reveal that the main assumptions cause increasingly negative effects when the embedding manifolds fail to accurately capture the complex, high-dimensional states.
> Specifically,
> applying regularization on the control solutions (as shown in Equation 3 in Section 2.1) can mitigate the inaccuracies of embedding manifolds.
> However, as the accuracy of the embedding manifolds decreases, the effectiveness of the control solution becomes more uncertain, thereby complicating the optimal control problem.
> The increased complexity in the optimal control problem makes the negative impact of violating the main assumptions more significant.
>
> We expand on this comparison by generating linear embedding subspaces that capture varying levels of accumulated variances, aiming to capture 99\%, 95\%, 90\%, and 85\% of the variances of the underlying states.
> As we lower the threshold for variance, the precision of these embedded subspaces decreases, thereby introducing more complexity into solving the optimal control problems (an extreme case is when the embedding manifolds are ground-truth manifolds for the underlying states, then a simply greedy solution of orthogonal projection is equivalent to the optimal control solution).
> The analysis indicates that the discrepancies in performance between the analytical solution and Pontryagin's Maximum Principle are minimal at higher variance levels (e.g., 99\%), but this difference becomes more pronounced at lower variance levels (e.g., 90\% and 85%).
> In such cases, the application of PMP, which functions without depending on oversimplified assumptions, provides significantly improved control solutions.
> The numerical validation shows that the main assumptions cause minimal negative effects when the embedding manifolds can accurately capture the high-dimensional states.
>
> Observe that the performance of PMP is slightly worse than that of the analytic solution when utilizing accuracy embeddings. This is due to the difficulties associated with solving PMP through optimization techniques. We have set the learning rate at 0.1 and the number of iterations at 3 for consistency.
>
> **Distilbert**
> |          | Base | 0.99%      | 0.95%      | 0.9%       | 0.85%      |
> |----------|------|------------|------------|------------|------------|
> | None     | 87.23| 85.88 / 86.52| 68.92 / **80.21**| 34.24 / **54.06**| 34.28 / **46.75**|
> | A2T      | 53.89| 61.75 / 60.87| 57.93 / **62.88**| 34.10 / **44.17**| 34.28 / **42.01**|
> | PSO      | 49.84| **54.33** / 52.80| 56.21 / **58.22**| 34.13 / **46.27**| 34.28 / **42.55**|
> | TextBugger| 24.73| **40.35** / 36.69| **43.89** / 42.50| 36.24 / **39.02**| 34.28 / **42.20**|
> | TextFooler| 24.69| **40.28** / 36.13| **49.05** / 46.69| 34.37 / **39.56**| 34.28 / **40.80**|
>
> **RoBERTaBase**
> |            | Base | 0.99%      | 0.95%      | 0.9%       | 0.85%      |
> |------------|------|------------|------------|------------|------------|
> | None       | 90.87| 90.10 / 90.59| 85.09 / **89.63**| 64.60 / **85.82**| 40.02 / **76.71**|
> | A2T        | 58.36| **63.82** / 62.19| 65.12 / **66.44**| 51.19 / **66.86**| 37.56 / **60.41**|
> | PSO        | 51.44| **54.36** / 52.98| **59.97** / 56.75| 52.55 / **59.18**| 37.91 / **59.07**|
> | TextBugger | 35.90| **43.03** / 40.53| 46.74 / 46.41| 38.72 / **47.36**| 34.84 / **42.43**|
> | TextFooler | 27.03| **37.18** / 33.47| **47.16** / 42.79| 41.40 / **46.60**| 35.76 / **45.49**|
>
> **RoBERTaLarge**
> |            | Base | 0.99%      | 0.95%      | 0.9%       | 0.85%      |
> |------------|------|------------|------------|------------|------------|
> | None       | 92.39| 91.98 / 92.11| 86.50 / **91.68**| 66.40 / **90.53**| 44.54 / **86.52**|
> | A2T        | 59.40| **64.64** / 63.15| **67.17** / 65.03| 54.67 / **64.99**| 41.54 / **61.94**|
> | PSO        | 52.15| **56.62** / 54.35| **62.14** / 55.51| 55.13 / **58.41**| 41.15 / **58.85**|
> | TextBugger | 33.72| **42.39** / 39.18| **47.48** / 41.59| 41.30 / **43.77**| 35.49 / **40.32**|
> | TextFooler | 26.43| **36.92** / 32.27| **48.79** / 37.14| **47.09** / 41.43| 40.47 / **41.97**|

---

> ### Author Response · Authors · 2024-02-12
> **Reponse to the choice of PD control**
>
> We have implemented the proposed control framework in all numerical experiments following the P-D control scheme.
> Further details on the implementation will be provided in Section 3.1.
> The reason why P-D outperforms P-I-D is mainly due to noise sensitivity and hyperparameter tuning.
>
> **Noise sensitivity:**
> The integral term has the potential to aggregate errors across multiple hidden layers, incorporating noise inherent in the embedding manifolds, as well as the distributional shift between the training and testing datasets.
> In scenarios where substantial noises are presented in each hidden layer, the integral component, dependent on the embedding manifold of accumulated past states, may lead to instability during model inference.
> Conversely, a Proportional-Derivative (PD) controller, lacking the integral component, tends to exhibit improved performance under such noisy conditions by not accumulating this noise.
>
> **Hyperparameter tuning:**
> In the realm of traditional PID control design, selecting the appropriate control gains, denoted as $K_p$, $K_i$, and $K_d$, for proportional, integral, and derivative controls respectively, presents a notable challenge.
> These gains are crucial for achieving a balance among the different types of controls.
> Typically, the calibration of these gains is empirically based, with the aim of optimizing the performance of PID control.
> Our method follows a similar strategy, determining the gains through experimentation with training data.
> Given that our hyperparameter searching space only contains $0$ and $0.5$ for each control gain,
> this results in the values $K_p = 0.5$, $K_d = 0.5$, and $K_I = 0$.
> A more principled method would entail adjusting these hyper-parameters through numerical optimization, treating these control gains as adjustable variables.
> The development of a more sophisticated strategy for fine-tuning the control gains will be studied for future exploration.

---

> ### Author Response · Authors · 2024-02-12
> **Reponse to the requested discussion on the computational wall time comparison**
>
> Here we present a detailed discussion of the computation overhead of the proposed PID control method.
> Specifically, we compare the computational wall time between the base model without any controls applied, the proposed analytic solution, and Pontryagin's maximum principle employed in the previous closed-loop control approach.
> As shown in this Table,
> across all four models,
> the computational wall time between the base model and the proposed analytic solution is comparable,
> the analytic solution only adds a small amount of wall time during inference.
> However, solving the PMP significantly adds to the computational wall time of the base model.
>
> In the following table, we calculate the computational wall time of four models on the SNLI testing dataset. Each number is recorded by averaging over 5 experiments and in units of second (s).
> Notice that OPT-1.3B has 1.3 billion model parameters, running the PMP on the OPT-1.3B model requires a significant amount of memory. With the same batch size across all experiments, it is not feasible to run PMP on this large model.
>
> |                 | Distilbert | RoBERTaBase | RoBERTaLarge | OPT    |
> |-----------------|------------|-------------|--------------|--------|
> | Base model      | 6.3751     | 11.7756     | 36.0178      | 123.5379|
> | Controlled model| 6.4221     | 11.8620     | 36.5051      | 124.1090|
> | PMP             | 62.2667    | 81.2795     | 263.7920     | N/A|

---

### Review · Reviewer_TdCP · 2024-02-12

**Summary Of Contributions:**

The authors propose correction mechanism using PID (Proportional-Integral-Derivative) control for robustness of pre-trained large language models.

**Audience:**

Yes

**Broader Impact Concerns:**

No broader impact concerns

**Claims And Evidence:**

Yes

**Requested Changes:**

Comments:

- The following few sentences are not entirely clear: “The methodology of constructing a self-healing process to improve the robustness of deep neural networks was initially introduced in Chen et al. (2020) and its subsequent work Chen et al. (2022). It leveraged a closed-loop control method to detect and correct potential errors applied to input data. This method belongs to a special case of P control in the proposed PID control framework, in which only the errors from the present states are corrected.” “It leveraged”… refers to Chen et al, I would understand?

- Crucially, Table 1 and Table 2 should indicate why these results are statistically significant (in many cases, the difference is only a few percentage points, so seeing a confidence interval would be crucial to understanding whether these results are significantly different from the baseline). The claims such as “In models trained using standard methods, the implementation of P-D control shows an average performance improvement of 5% over the P control scheme. In robustly trained models, this improvement is nearly 3%.“ should be backed up by some confidence intervals. Furthermore, it seems from Table 2 that the results do not show the consistent winner method.

- The ablation study excludes certain parts of the PID controller. It is, however, unclear whether P-baseline exactly corresponds to Chen et al (2020) or not. This is especially important given that at the end of page 2, the authors discuss computational inefficiency of  the optimal control solution by Chen et al (2020, 2022) which implies that they are not just the P-ablation of the current method. Besides, it would be great to see if one can compare it with both Chen et al (2020) and Chen et al (2022)

- It is important to discuss the wider perspective of the proposed method on the trustworthy ML and how it complements the research on adversarial attacks. Can the same method be used for vision problems, for example? How does it complement the existing trustworthy ML literature? Perhaps, adding more information into the Related Work section could help.

- It would be also good to see if it is possible to empirically calculate the error in Section 2.3 and demonstrate it through forward propagation?

**Strengths And Weaknesses:**

Pros:

- Interesting idea, which extends the use of PID control introduced by Chen et al (2020) and Chen et al (2022)

- The method appears to address the important questions of robustness to the community

- I checked the derivations and they look correct to me.

Cons:

Clarity and experimental evidence needs to be improved (see below), due to the experimental evidence issues I answer 'no' on the claims and evidence question below.

---

> ### Author Response · Authors · 2024-02-13
> **Clarification on Ambiguous Sentences**
>
> Thank you for your interest in this work and insightful comments. Here we answer your first question regarding these ambiguous sentences.
>
> In the sentence "It leveraged a closed-loop control method ...",
> it refers to Chen et al.
> The aim of these sentences is to highlight how the proposed PID control formulation generalizes the previous closed-loop control approach.
> Further clarification on these sentences is provided in the following.
>
> Previous studies, specifically those by Chen et al. (2020) and Chen et al. (2022), have developed a closed-loop control approach aimed at improving the robustness of deep neural networks.
> This approach formulates the robustness issue of deep neural networks as a trajectory optimization problem.
> Here, the reference trajectory is constructed as a sequence of embedding manifolds of the hidden states.
> In the process of online inference, these embedding manifolds are used to measure the distance from the actual states to the embedding manifolds, and an optimal control problem is formulated to minimize this distance.
>
> The proposed PID-control formulation generalizes the previous closed-loop method by integrating additional integral and derivative control mechanisms.
> As shown in Equation 3 from the submitted manuscript, the central element of the optimal control objective function, the running loss, is composed of four parts: the normed distances from the controlled state to the state's embedding manifolds, the integration of past states, the state derivative, and a term for control regularization.
> The variables $K_p$, $K_I$, and $K_d$ represent the control gains for proportional, integral, and derivative controls, respectively, with Equation 3 assigning values of 0.5 to each.
> The original closed-loop control method can be constructed by setting $K_p$ to 0.5 and both $K_I$ and $K_d$ to 0.
> The PID control formulation broadens the scope of the prior closed-loop control method, offering more flexible control options.
> For example, derivative controls become crucial in scenarios where states undergo significant variations over time or across different layers.
> Conversely, in situations where state dynamics evolve smoothly, integral controls can be more effective compared to both proportional and derivative control schemes.
>
> In this study, we derive an analytical solution for fast inference, addressing a notable challenge encountered in prior works.
> When the PID control is generalized to the previous closed-loop control by setting $K_p=0.5$, $K_I=K_d=0$,
> our control solution still differs from that of previous studies due to our employment of an analytical solution, as opposed to Pontryagin's Maximum Principle utilized before.
> Additionally, following your third request for revisions, we present a comparative analysis of the PID control using our analytical approach against the previous closed-loop control method that solves Pontryagon's Maximum principle during online inference.

---

> ### Author Response · Authors · 2024-02-13
> **Reponse to the requested change of analyzing confidence intervals**
>
> In Table 2, we conduct an ablation study to choose the optimal control scheme,
> rather than compare the proposed PID control with any baseline methods.
> Table 2 considers potential candidates including
> Proportional (P) control,
> Proportional-Integral (P-I) control,
> Proportional-Derivative (P-D) control,
> and Proportional-Integral-Derivative (P-I-D) control.
> All these four candidate approaches are constructed by the proposed PID control formulation defined in Equations 2 and 3.
> Although the generalized Proportional (P) control has the same objective function as the previous closed-loop control approach in Chen et al. (2020) and Chen et al. (2022),
> the numerical results in Table 2 are obtained from the derived analytic solution,
> which differ from the previous approaches that rely on an iterative solver during online inference.
> Therefore, the P control scheme in Table 2 should not be considered as a baseline method.
>
> The Proportional-Derivative (P-D) control outperforms other control schemes in most tasks,
> we choose this P-D control in all experiments that involve comparing with baseline methods (Radar plot 2, Table 1, Table 3, and Table 4).
> Radar plot 2 and Table 1 show the performance improvement of model robustness in the ANLI dataset and against various adversarial attacks, respectively.
> More detailed numerical results regarding Radar plot 2 are shown in Tables 3 and 4 in Appendix C.
>
> The following Table shows the mean and confidence interval to better understand the numerical results.
> In Table 2,
> the mean of employing the Proportional-Derivative (P-D) control over the Proportional (P) control is 2.35%, with a 95% confidence interval of 1.677% to 3.0121%.
> This validates our choice of P-D control.
> Table 1 shows the robustness improvement on the ANLI dataset.
> The ANLI dataset is an adversarial dataset in which the testing data are significantly different from the training dataset.
> The proposed control method leads to 1.0783% in the mean of performance improvement, and a 95% confidence interval of 0.0564% to 2.1004%.
> Moreover, in standard adversarial attacks,
> the performance improvement is more significant.
> As shown in Tables 3 and 4,
> the means of performance improvement are 5.9577% and 5.0031%,
> with a 95% confidence interval of 4.5722% to 7.3432% and 3.9450% to 6.0613%.
>
> **Mean and 95% Confidence Interval**
> | Table    | Mean  | Lower | Upper |
> |----------|-------|-------|-------|
> | Table 1  | 1.0783| 0.0564| 2.1004|
> | Table 2  | 2.35  | 1.677 | 3.0121|
> | Table 3  | 5.9577| 4.5722| 7.3432|
> | Table 4  | 5.0031| 3.9450| 6.0613|

---

> ### Author Response · Authors · 2024-02-13
> **Reponse to the requested change of comparison with baseline method**
>
> The Proportional (P) control baseline does not exactly correspond to the previous works in the ablation study.
> The ablation study aims to find the optimal control scheme from Proportional (P) control,
> Proportional-Integral (P-I) control,
> Proportional-Derivative (P-D) control,
> and Proportional-Integral-Derivative (P-I-D) control.
> These control problems are solved by the derived analytic solution, rather than solving Pontryagin's Maximum Principle as done in previous works.
>
> The following Table shows the performance comparison between the closed-loop control in Chen et al (2020) and Chen et al (2022) and the PID control proposed in this work.
> As can be seen,
> the PID control outperforms the baseline method in most of the evaluation tasks.
> The mean of the performance improvement of the proposed PID control over the baseline method is 3.1643%,
> with a 95% confidence interval of 1.9751% to 4.3535%.
>
> **Performance comparison. Each entry: Chen et al (2020) and Chen et al (2022) / This work**
>
> **Distilbert:**
>
> | Attack     | Standard training | Adversarial training |
> |------------|------------------------------|---------------------------------|
> | None       | **87.04** / 85.88                | **86.54** / 85.39                   |
> | A2T        | 57.77 / **61.75**                | 72.23 / 72.36                   |
> | PSO        | 51.41 / **54.33**                | 54.03 / **57.46**                   |
> | TextBugger | 32.20 / **40.35**                | 34.91 / **41.11**                   |
> | TextFooler | 30.87 / **40.28**                | 35.93 / **44.03**                   |
>
> **RoBERTaBase:**
>
> | Attack     | Standard training | Adversarial training |
> |------------|-------------------------------|----------------------------------|
> | None       | **91.11** / 90.10                 | 90.53 / 89.80                    |
> | A2T        | 61.04 / **63.82**                 | 76.73 / 76.97                    |
> | PSO        | 52.56 / **54.36**                 | 54.51 / **56.08**                    |
> | TextBugger | 38.51 / **43.03**                 | 39.54 / **43.72**                    |
> | TextFooler | 30.67 / **37.18**                 | 32.04 / **39.28**                    |
>
> **RoBERTaLarge:**
>
> | Attack     | Standard training | Adversarial training |
> |------------|--------------------------------|-----------------------------------|
> | None       | 92.46 / 91.98                  | 91.86 / 91.36                     |
> | A2T        | 62.80 / **64.64**                  | 81.55 / 81.64                     |
> | PSO        | 53.87 / **56.62**                  | 56.38 / **59.68**                     |
> | TextBugger | 39.70 / **42.39**                  | 40.71 / **44.91**                     |
> | TextFooler | 31.49 / **36.92**                  | 34.11 / **42.61**                     |
>
> Additionally, in the following Table, we present a comparison of the computational wall time between our analytic solution and the iterative solver previously utilized by Chen et al. (2020) and Chen et al. (2022).
> It is evident that the computational wall time required for our proposed analytic control solution is on par with that of the baseline models' inference times.
> In contrast, the computational wall time is considerably extended when using the prior closed-loop control approach, which proves to be excessively expensive, especially in the context of large-scale models, such as those with 1.3 billion parameters like OPT.
>
> **Wall Time (s) of 10,000 Test Samples (averaged over 5 experiments**
>
> | Model            | Distilbert | RoBERTaBase | RoBERTaLarge | OPT    |
> |------------------|------------|-------------|--------------|--------|
> | Base model       | 6.3751     | 11.7756     | 36.0178      | 123.5379 |
> | Analytic control | 6.4221     | 11.8620     | 36.5051      | 124.1090 |
> | Chen et al (2020)| 62.2667    | 81.2795     | 263.7920     | 757.0649 |

---

> ### Author Response · Authors · 2024-02-14
> **Reponse to the discussion for the wider perspective and more details**
>
> **The wider perspective of the proposed method on the trustworthy ML:**
>
> The presented PID control approach generalizes previous closed-loop control approaches with additional integral and derivative controllers.
> This development leads to more flexible control schemes,
> derivative controllers are more effective when the underlying states change rapidly,
> integral controllers play more significant roles when lower-dimensional embedding structures can be constructed in accumulated states.
> Such flexibility in control design broadens the applicability of the control framework across a variety of trustworthy ML applications.
>
> This work paves the way for the development of robust large language models.
> Presently, many large language models face challenges related to trustworthiness, including biases against minority groups in natural language generation tasks.
> In principle, by constructing state embedding manifolds that capture desired model behaviors, the PID control framework can be employed to adjust any unwanted behaviors in the model.
> This idea is similar to prompt engineering techniques used to modify input strings for achieving specific outcomes from models.
> These avenues will be explored further in future research.
>
> **How this complements the research on adversarial attacks:**
>
> The PID control framework leads to a new method to generate adversarial attacks.
> In the current work, the aim is to improve model robustness by minimizing the objective function defined in Equation 2.
> On the contrary,
> maximizing this loss w.r.t. some input perturbation is equivalent to generating adversarial examples.
> This can be an optimal control-based adversarial attack algorithm.
>
> **Can the same method be used for vision problems:**
>
> This method is applicable to computer vision problems.
> Typically, in deep convolutional neural networks, both the input and hidden states lie in extremely high-dimensional spaces, where the embedding manifolds for these states tend to exist. This assumption aligns with the "manifold hypothesis," which is based on the characteristics of real-world image data, and is further supported by empirical evidence as demonstrated in the studies by Chen et al. (2020) and Chen et al. (2022).
> Once the embedding manifolds for both input and hidden states are constructed, it becomes possible to formulate the optimal control objective function as outlined in Equations 2 and 3. By aiming to minimize this objective function, there is a potential to significantly improve the robustness of the model.
>
> **How does this complement the existing trustworthy ML literature:**
>
> The current body of research on trustworthy machine learning predominantly emphasizes adversarial training, which leads to two significant challenges.
> Firstly, the process of modifying model parameters with adversarial examples demands extensive computational resources.
> In the context of natural language processing tasks, identifying an adversarial example typically entails solving a combinatorial optimization problem,
> which suffers from an exponential growth in the number of feasible solutions as the size of the problem increases.
> Secondly, adversarial training's efficacy diminishes in the face of unexpected adversarial attacks.
> This shortcoming is especially evident in the real-world application of large language models, where predicting potential adversarial attacks beforehand is unfeasible.
> The suggested PID control framework is designed to overcome these challenges by offering two key advantages: 1) It does not significantly increase the inference time when compared to the base model, and 2) It leverages the embedding structure of unperturbed states, making it robust against various adversarial attack algorithms.

---

> ### Author Response · Authors · 2024-02-14
> **Reponse to the requested change for error computation**
>
> We provide the details of the error computation outlined in Theorem 4 of Section 2.3.
> Our objective is to demonstrate that the accuracy of the error computation specified in Theorem 4 diminishes with the addition of more layers to the language model.
> This decrease in accuracy is due to the assumptions of linearity and orthogonality.
> According to these assumptions, the transformations applied to each layer of a language model merely rotate the hidden state without altering its magnitude.
> However, as the model incorporates more of these layer-wise transformations, the accuracy of these assumptions starts to decrease.
>
>
> The table below presents the calculation of the absolute difference between the actual error and the error estimate as per Theorem 4.
> It is evident that with all types of adversarial perturbations (A2T, PSO, TextBugger, and TextFooler), the increase in the number of layers within the language model (with 6 layers representing Distilbert, 12 layers symbolizing RoBERTaBase, and 24 layers signifying RoBERTaLarge) leads to a rise in the absolute error.
> This indicates a decline in the precision of the error estimation.
>
> **Absolute error between true error and the error estimation in Theorem 4**
>
> |               | 6 Layers | 12 Layers | 24 Layers |
> |---------------|----------|-----------|-----------|
> | A2T           | 3.2189   | 4.3062    | 5.1566    |
> | PSO           | 1.9156   | 2.6087    | 3.3047    |
> | TextBugger    | 3.2189   | 4.3062    | 5.1566    |
> | TextFooler    | 3.1348   | 4.2894    | 5.2915    |
>
> Furthermore, this validates the quantitative findings depicted in Radar Plot 2 and illustrated in Tables 3 and 4.
> To elaborate, using the SNLI dataset across five evaluation tasks (non-perturbed, A2T, PSO, TextBugger, and TextFooler), the controlled Distilbert model, which contains 6 layers, demonstrates an average performance improvement of 11.5275 with a 95% confidence interval ranging from 2.4738 to 20.5811.
> Conversely, when the number of layers is increased, the performance gains in the controlled RoBERTaLarge model, which has 24 layers, are modest with an average performance of 7.4175 and a 95% confidence interval between 2.8028 to 12.0321.

---

> ### Comment · Reviewer_TdCP · 2024-03-03
>
> Many thanks to the authors for thoroughly addressing the comments. I am happy with the explanation around my points.
>
> A couple of outstanding points I would like to raise:
> - in relation to the current large language models (Reviewer Tx49), it would be important to state explicitly in the paper that although the framework would allow such models (as shown in the experiments with the OPT model), the experiments focus on smaller models due to large computational resources required and therefore, only OPT model has been considered amongst the larger models.
> -  in relation to the question from Reviewer sddg, would the authors be able to further present the notion of undesirable perturbations  in a formal way to improve clarity?

---

> > ### Author Response · Authors · 2024-03-04
> > **The notion of undesirable perturbations**
> >
> > Thank you for your positive feedback. We will add the suggested comments about the experiment setting and the following notion of undesirable perturbations in the revised manuscript.
> >
> > We create embedding manifolds from clean training data to model how states behave without external perturbations. Any state that lies outside of these embedding manifolds can be considered as the state of perturbed input. This error can be detected and corrected by the PID control.
> > Let's define "x" as the input data, and "\overline{x}" as the perturbed data, where "$\overline{x}$" equals x plus z, with z symbolizing a general perturbation. This perturbation, z, can be decomposed into a direct sum of two orthogonal components: $z^{\parallel}$, a perturbation within the data manifold, and $z^{\perp}$, a perturbation orthogonal to the data manifold. Our PID control focuses on identifying and correcting the orthogonal perturbation, $z^{\perp}$. Thus, within the context of embedding manifolds, undesirable perturbations are defined as any perturbation that has a non-zero orthogonal component, $z^{\perp}$, relative to the embedding manifold.

---

### Review · Reviewer_sddg · 2024-02-19

**Summary Of Contributions:**

The paper proposes a PID control framework to improve the robustness of LLMs against input _embeddings_ perturbations. The key ideas are noticing the parallels between neural network depth and discrete dynamical systems, allowing formulation of robustness as a trajopt problem.
This allows designing PID controller for each layer to monitor errors between desired & actual internal states. Linear embedding manifolds for the PID states are constructed by leveraging the geometric properties of training data. The analytical solution is derived under a big assumption of linear orthogonality of transformations.
The approach is evaluated on NLI tasks using adversarial attacks to demonstrate improved robustness.

**Audience:**

Yes

**Claims And Evidence:**

No

**Requested Changes:**

Given the paper's motivations, would be good to clarify how the paper's assumptions connect to real world threat models.

**Strengths And Weaknesses:**

The approach is computationally efficient and avoids expensive on-line optimization while also avoiding compromising on accuracy. Lots of theoretical analysis.

However, while it proposes a promising approach for an artificial embedding perturbation problem, I don't understand how it's applicable for any real world scenario for enhancing NLP robustness. The proposed approach assumptions seem disconnected from any practical adversarial attacks and the it's overall hard to figure out what exactly is being measured in the reported experimental results.
To go into slightly more detail:
- The method assumes perturbations and threats exist directly in the continuous embedding space of neural networks. However, most real-world adversarial attacks on LLMs would be expected to manipulate discrete tokens or characters in the input text? This calls into question the practical utility of making models robust to embedding perturbations. If the adversary has access to manipulate the network activations, there might be bigger security issues with the deployment. The threat model seems quite artificial.
- It's unclear that any intentionally crafted input level perturbations would truly manifest as embedding noise that could be corrected via PID control theory.
- Embeddings themselves don't have any ground truth labels, so it's unclear what "undesirable" perturbation even means in this context.

Other issues are the obviously big simplifying assumption about linear orthogonal transformations that enables the performance claims. I don't understand when they would practically hold.

---

> ### Author Response · Authors · 2024-02-19
> **Responses to perturbation setting, evidence on the effectiveness of error correction via PID control**
>
> Thank you for your insightful comments. Below are our responses:
>
> **Clarification on the perturbation setting:**
> In this work, we consider word/token level adversarial attacks that manipulate discrete tokens or characters in the input text.
> We verify this from both empirical and theoretical perspectives.
>
> Empirical verification:
> In the numerical experiments,
> we consider a range of adversarial attacks, including A2T, PSO, TextBugger, and TextFooler.
> These adversarial attacks aim to cause misclassification by modifying the tokens of the input string while maintaining the same semantic meaning.
> As shown in Radar Plot 2 and Tables 3 and 4, the PID control framework consistently improves the model robustness of these word-token level adversarial attacks across all models and against all perturbations.
>
> Theoretical verification:
> Theorem 2.3 outlines how errors in state computations at any given time step are influenced by input perturbations represented by $z$, despite these perturbations existing within the continuous domain of $\mathbb{R}^d$,
> this setting fits real-world adversarial attacks on LLMs, which involve modifying discrete elements, such as tokens or characters, in the input text.
> The act of modifying a word or substring through an adversarial attack leads to a discrepancy between the embedding sequences of the original and modified input tokens, manifesting as the perturbation vector $z$ within the input embedding space.
> Specifically, the embedding manifolds, derived from unperturbed training data, capture the structure of this data in a lower-dimensional subspace.
> Adversarial examples, meanwhile, are designed to be semantically similar to the original input yet induce a marked divergence in the embedding space during the model's forward propagation.
> Under these circumstances, the difference between the embedding sequences of the original input and the adversarial example can be quantified and adjusted within the PID control framework, as detailed in Theorem 2.3.
>
> **Evidence on error correction via PID control:**
> Here we provide evidence that the PID control framework can improve model robustness against adversarial examples.
> As detailed in Theorem 4 in Section 2.3,
> the working principle of the PID control framework is based on the two facts:
> 1. There exists an embedding structure in the state at every layer.
> 2. The sequence of states from adversarially perturbed input deviates from the true embedding structures.
>
> The following Table verifies the existence of lower-dimensional embedding subspaces. With the OPT-1.3 B LLM (only the first 6 layers are shown), the dimensions of proportional, integral, and derivative embedding subspaces are presented. As can be seen, the dimension of proportional embedding is around 350 on average in a 2048 dimensional space, integral and derivative embedding subspaces also show low dimensions compared with the full space (2048).
> |              | Layer 1   | Layer 2   | Layer 3   | Layer 4   | Layer 5   | Layer 6   |
> |--------------|-----------|-----------|-----------|-----------|-----------|-----------|
> | Proportional | 191 / 2048| 148 / 2048| 224 / 2048| 337 / 2048| 451 / 2048| 549 / 2048|
> | Integral     | 191 / 2048| 181 / 2048| 180 / 2048| 212 / 2048| 253 / 2048| 298 / 2048|
> | Derivative   | 191 / 2048| 355 / 2048| 1001 / 2048| 1352 / 2048| 1494 / 2048| 1535 / 2048|
>
> The following Table presents the error (measured in 2-norm) detected by the combination of P, I, and D embedding subspaces. As can be seen, the perturbation aims to amplify the error as propagated into deeper layers, and the embedding subspaces can effectively detect these errors at all layers.
> | OPT-1.3B (first 6 layers) | Layer 1 | Layer 2 | Layer 3 | Layer 4 | Layer 5 | Layer 6 |
> |---------------------------|---------|---------|---------|---------|---------|---------|
> | OPT                       | 1.8237  | 9.5877  | 6.9810  | 6.3207  | 7.5278  | 16.6526 |
>
> **Clarification on the term undesirable perturbations:**
> The term "undesirable perturbation" refers to perturbations that lead to abnormal embedding sequences during the model's forward propagation.
> We address the issue of robustness by framing it as a trajectory optimization problem, with the goal of generating control signals that ensure embedding sequences follow estimated true embedding sequences at every time step.
> These approximated true sequences are represented by a series of embedding manifolds based on unperturbed training data.
> Adversarial perturbations aim to amplify the noise within the model's hidden states throughout forward propagation, resulting in the embedding sequence deviating from these true sequences.
> Our method focuses on identifying such deviations across all layers of the model and correcting them through the application of control signals.

---

> ### Author Response · Authors · 2024-02-19
> **Discussion on the main assumptions**
>
> The two foundational assumptions underlying the analytic solution's derivation are not typically valid in real-world applications of conventional deep learning networks.
> Despite this, these assumptions are crucial for deriving the analytic solution that allows for fast inference, which is a major challenge in existing works.
> Here we discuss the negative impact of violating the assumptions made to derive the analytic solution.
> Our analysis uncovers that the negative impact on model performance only emerges when the embedding manifolds are unable to precisely represent the high-dimensional states.
> Conversely, when these manifolds are constructed accurately, the performance under the assumptions of linearity and orthogonality is found to be on par with that of Pontryagin's maximum principle, an iterative approach that does not rely on these assumptions, as detailed in Section 2.1.
>
> The application of regularization on control solutions can lessen the errors from the embedding manifolds.
> Yet, as the accuracy of these manifolds decreases, there's a need for more degree of regularization, which in turn, adds complexity to the optimal control problem.
> This increase in complexity amplifies the adverse effects of deviating from the main assumptions.
>
> Our evaluation includes comparing the efficacy of our proposed analytic solution with that of Pontryagin's Maximum Principle, an iterative method that doesn't rely on extra assumptions.
> Pontryagin's Maximum Principle provides the necessary conditions for an optimal control solution, typically offering a robust approximation of such solutions.
> We expand on this comparison by generating linear embedding spaces that aim to account for 99%, 95%, 90%, and 85% of the variances in the states, with each lower variance threshold indicating a decrease in the precision of these subspaces and, consequently, presenting increased difficulties in addressing optimal control problems.
> The comparative analysis, detailed in the following Table, assesses three language models (Distilbert, RoBERTaBase, and RoBERTaLarge) across five evaluation tasks.
> These include a standard test without any disturbances and four adversarial challenges: A2T, PSO, TextBugger, and TextFooler.
> Findings indicate that the disparity in outcomes between the analytic solution and Pontryagin's Maximum Principle becomes negligible when the embedding subspaces are highly accurate (e.g., 99% variance captured) but grow significantly with inaccurate embedding subspaces (e.g., 90% and 85% variances).
> In such cases, the application of Pontryagin's Maximum Principle, which is not constrained by simplifying assumptions, results in better control solutions.
>
> **Distilbert:**
> |            | Base  | 0.99%            | 0.95%            | 0.9%             | 0.85%            |
> |------------|-------|------------------|------------------|------------------|------------------|
> | None       | 87.23 | 85.88 / 86.52    | 68.92 / **80.21**    | 34.24 / **54.06**    | 34.28 / **46.75**    |
> | A2T        | 53.89 | 61.75 / 60.87    | 57.93 / **62.88**    | 34.10 / **44.17**    | 34.28 / **42.01**    |
> | PSO        | 49.84 | **54.33** / 52.80    | 56.21 / **58.22**    | 34.13 / **46.27**    | 34.28 / **42.55**    |
> | TextBugger | 24.73 | **40.35** / 36.69    | **43.89** / 42.50    | 36.24 / **39.02**    | 34.28 / **42.20**    |
> | TextFooler | 24.69 | **40.28** / 36.13    | **49.05** / 46.69    | 34.37 / **39.56**    | 34.28 / **40.80**    |
>
> **RoBERTaBase:**
> |            | Base  | 0.99%          | 0.95%          | 0.9%           | 0.85%          |
> |------------|-------|----------------|----------------|----------------|----------------|
> | None       | 90.87 | 90.10 / 90.59  | 85.09 / **89.63**  | 64.60 / **85.82**  | 40.02 / **76.71**  |
> | A2T        | 58.36 | **63.82** / 62.19  | 65.12 / **66.44**  | 51.19 / **66.86**  | 37.56 / **60.41**  |
> | PSO        | 51.44 | **54.36** / 52.98  | **59.97** / 56.75  | 52.55 / **59.18**  | 37.91 / **59.07**  |
> | TextBugger | 35.90 | **43.03** / 40.53  | 46.74 / 46.41  | 38.72 / **47.36**  | 34.84 / **42.43**  |
> | TextFooler | 27.03 | **37.18** / 33.47  | **47.16** / 42.79  | 41.40 / **46.60**  | 35.76 / **45.49**  |
>
> **RoBERTaLarge:**
> |            | Base  | 0.99%          | 0.95%          | 0.9%           | 0.85%          |
> |------------|-------|----------------|----------------|----------------|----------------|
> | None       | 92.39 | 91.98 / 92.11  | 86.50 / **91.68**  | 66.40 / **90.53**  | 44.54 / **86.52**  |
> | A2T        | 59.40 | **64.64** / 63.15  | **67.17** / 65.03  | 54.67 / **64.99**  | 41.54 / **61.94**  |
> | PSO        | 52.15 | **56.62** / 54.35  | **62.14** / 55.51  | 55.13 / **58.41**  | 41.15 / **58.85**  |
> | TextBugger | 33.72 | **42.39** / 39.18  | **47.48** / 41.59  | 41.30 / **43.77**  | 35.49 / **40.32**  |
> | TextFooler | 26.43 | **36.92** / 32.27  | **48.79** / 37.14  | **47.09** / 41.43  | 40.47 / **41.97**  |

---

### Author Response · Authors · 2024-02-19
**Discussion on the two main assumptions**

Here we discuss the two main assumptions, which are common concerns of all reviewers.
This post is a summary of our early discussion with AE.

Here we discuss the negative impact of violating the assumptions made to derive the analytic solution.
Through empirical evaluations, we highlight how the main assumptions have increasingly adverse effects, especially when the embedding manifolds fail to accurately capture the complex, high-dimensional states.
More specifically,
applying regularization on control solutions can mitigate these inaccuracies.
However, as the precision of the embedding manifolds decreases, a greater degree of regularization is required, thereby complicating the optimal control problems.
The increased complexity in the optimal control problem makes the negative impact of violating the main assumptions more significant.

Our validation approach involves a performance comparison between the proposed analytic solution and the implementation of Pontryagin's Maximum Principle, an iterative solver that operates without the need for additional assumptions.
Pontryagin's Maximum Principle provides the necessary conditions for an optimal control solution, typically offering a robust approximation of such solutions.
We further elaborate this comparison by creating linear embedding subspaces with varying thresholds for accumulated variances, specifically aiming to capture 99%, 95\%, 90\%, and 85\% of the variances in the underlying states.
As the variance threshold is lowered, the accuracy of these embedding subspaces decreases, thus posing greater challenges in solving optimal control problems.
The performance comparison, detailed in the following table, includes three large language models, namely Distilbert, RoBERTaBase, and RoBERTaLarge, across five evaluation tasks.
These tasks include a standard scenario with no perturbation and four adversarial attacks: A2T, PSO, TextBugger, and TextFooler.
The results reveal that while the performance difference between the analytic solution and Pontryagin's Maximum Principle is negligible at higher accuracy levels (e.g., 99\% variance), the scenario changes significantly at lower accuracies (e.g., 90\% and 85\$ variances).
In these instances, employing Pontryagin's Maximum Principle, which operates independently of simplifying assumptions, yields noticeably better control solutions.

Observe that the performance of PMP is slightly worse than that of the analytic solution when utilizing accuracy embeddings.
This is due to the difficulties associated with solving PMP through optimization techniques. We have set the learning rate at 0.1 and the number of iterations at 3 for consistency.

**Distilbert:**
|            | Base  | 0.99%            | 0.95%            | 0.9%             | 0.85%            |
|------------|-------|------------------|------------------|------------------|------------------|
| None       | 87.23 | 85.88 / 86.52    | 68.92 / **80.21**    | 34.24 / **54.06**    | 34.28 / **46.75**    |
| A2T        | 53.89 | 61.75 / 60.87    | 57.93 / **62.88**    | 34.10 / **44.17**    | 34.28 / **42.01**    |
| PSO        | 49.84 | **54.33** / 52.80    | 56.21 / **58.22**    | 34.13 / **46.27**    | 34.28 / **42.55**    |
| TextBugger | 24.73 | **40.35** / 36.69    | **43.89** / 42.50    | 36.24 / **39.02**    | 34.28 / **42.20**    |
| TextFooler | 24.69 | **40.28** / 36.13    | **49.05** / 46.69    | 34.37 / **39.56**    | 34.28 / **40.80**    |

**RoBERTaBase:**
|            | Base  | 0.99%          | 0.95%          | 0.9%           | 0.85%          |
|------------|-------|----------------|----------------|----------------|----------------|
| None       | 90.87 | 90.10 / 90.59  | 85.09 / **89.63**  | 64.60 / **85.82**  | 40.02 / **76.71**  |
| A2T        | 58.36 | **63.82** / 62.19  | 65.12 / **66.44**  | 51.19 / **66.86**  | 37.56 / **60.41**  |
| PSO        | 51.44 | **54.36** / 52.98  | **59.97** / 56.75  | 52.55 / **59.18**  | 37.91 / **59.07**  |
| TextBugger | 35.90 | **43.03** / 40.53  | 46.74 / 46.41  | 38.72 / **47.36**  | 34.84 / **42.43**  |
| TextFooler | 27.03 | **37.18** / 33.47  | **47.16** / 42.79  | 41.40 / **46.60**  | 35.76 / **45.49**  |

**RoBERTaLarge:**
|            | Base  | 0.99%          | 0.95%          | 0.9%           | 0.85%          |
|------------|-------|----------------|----------------|----------------|----------------|
| None       | 92.39 | 91.98 / 92.11  | 86.50 / **91.68**  | 66.40 / **90.53**  | 44.54 / **86.52**  |
| A2T        | 59.40 | **64.64** / 63.15  | **67.17** / 65.03  | 54.67 / **64.99**  | 41.54 / **61.94**  |
| PSO        | 52.15 | **56.62** / 54.35  | **62.14** / 55.51  | 55.13 / **58.41**  | 41.15 / **58.85**  |
| TextBugger | 33.72 | **42.39** / 39.18  | **47.48** / 41.59  | 41.30 / **43.77**  | 35.49 / **40.32**  |
| TextFooler | 26.43 | **36.92** / 32.27  | **48.79** / 37.14  | **47.09** / 41.43  | 40.47 / **41.97**  |

---

### Author Response · Authors · 2024-02-19
**Details on the description of optimal control to deep learning**

Here we delve deeper into the formulation of optimal control. This content summarizes our preliminary conversations with the EA, and it was mutually decided to share this information publicly for further details.

**An Optimal Control Framework**
We start with a description of the dynamical system approach to machine learning. In the dynamical system framework, we consider the input $\mathbf{x}_0$ as the initial condition of a system of difference equations,

$$
\mathbf{x}_{t+1} = F_t(\mathbf{x}_t + \pi_t(\mathbf{x}_t), \boldsymbol\theta_t),
$$

where $F_t$  represents a time-varying difference equation, $\pi_t: \mathbb{R}^d \rightarrow \mathbb{R}^d$ is a feedback controller that maps the current state $\mathbf{x}_t$ to a control action.
The goal of optimal control is to design these feedback controllers such that some objectives are satisfied. This can be represented as the objective function defined in Eq. 2 in Section 2.
In general, objectives in the real world can be structured as terminal and running loss functions.
Therefore, optimizing such an objective function aligns with achieving a real-world goal.
Take the development of autonomous vehicles as an example.
This involves guiding the vehicle to a destination along a specific route.
This challenge can be approached as an optimal control problem, where reaching the destination is expressed as a terminal loss function, and the deviation from the planned route is captured through a running loss function.

The essential task of supervised learning is to approximate some function
\begin{equation*}
F: \mathcal{X} \rightarrow \mathcal{Y},
{\rm where}
F = F_{T-1} \circ F_{T-2} \circ \cdots \circ F_1 \circ F_0,
\end{equation*}
which maps inputs in $\mathcal{X} \in \mathbb{R}^d$ (e.g. images, natural language sequences) to labels in $\mathcal{Y}$ (categories, numerical predictions).
The objective of developing robust deep neural networks can be formulated within an optimal control framework.
Here, the aim is to minimize the discrepancy between the model's predictions and the actual labels through a terminal loss function by implementing feedback controllers.
However, this ideal scenario is challenged by the practical limitation that true labels are unavailable during the model's inference phase.
As a consequence, our focus shifts towards developing robust model predictions through the development of running loss functions.
In this work, we introduce a running loss function (defined as Eq. 3 in Section 2.1) that assesses the state of control at each timestep $t$.
This running loss function calculates a loss by measuring the difference between the controlled state and certain embedding manifolds.
These manifolds capture the structural, integrative, and derivative aspects of state embeddings.

**Details on PID control.**
The Proportional-Integral-Derivative (PID) control is a feedback control mechanism widely used in industrial control systems and a variety of other applications requiring continuously modulated control.
A PID controller continuously calculates an error value as the difference between a desired setpoint and a measured process variable and applies a correction based on proportional, integral, and derivative terms,
aiming to minimize the error by adjustment of a control variable.

For each of the P, I, and D control components,
the proportional term produces an output that is proportional to the current error value.
The proportional response can be adjusted by multiplying the error by a proportional gain.
The integral term is concerned with the accumulation of past errors.
It seeks to eliminate the residual steady-state error that occurs with a pure proportional controller by integrating the error over time and then multiplying by the integral gain.
This action accelerates the movement of the process toward the setpoint.
The derivative term predicts system behavior and thus can improve the stability and speed of the system response.
It is calculated by determining the slope of the error over time and multiplying this rate of change by the derivative gain.
Derivative control helps to dampen the system response and prevent overshooting the setpoint.
The performance of a PID controller can dramatically affect the efficiency, stability, and safety of dynamic systems.
Properly tuned PID controllers are crucial for achieving desired levels of system performance.

In this work, we extend the traditional closed-loop control strategy into the domain of PID control.
To achieve this, we define three embedding manifolds through three unique surjective functions:
$f_t^P: \mathbb{R}^d \rightarrow \mathbb{R}^{(d - r)}$ for the state,
$f_t^I: \mathbb{R}^d \rightarrow \mathbb{R}^{(d - r)}$ for the integration of past states,
and $f_t^D: \mathbb{R}^d \rightarrow \mathbb{R}^{(d - r)}$ for the state's derivative.
The running loss is then broken down into three parts, each evaluating the controlled state from a different perspective.

---

> ### Comment · Action_Editor_VLWp · 2024-02-19
> **Content based on preliminary AE comments**
>
> Just to clarify for the reviewers, I had two comments to the authors that I decided to share after my initial read to give the authors maximal time to response. The first comment was that I thought it might be nice for the TMLR audience to give a nice intro to PID control. The second comment (see thread below) was to provide some measure or quantification to get a sense of whether and with which severity the method breaks down in practice if the theoretical assumptions required for the math. analysis and derivation break down. We decided to share this once the initial reviews are in to not bias the reviews.

---

### Decision · Action_Editor_VLWp · 2024-03-27

**Recommendation:** Accept as is

**Comment:**

While reviewers raised some concerns in their initial reviews, some of which I shared based on a first read of the paper, these concerns were addressed with the updated manuscript (added experiments, expanded writing, clarifications) and authors' responses. All reviewers have recommended a 'leaning accept' and I agree that all major open issues were addressed by the authors and the paper is ready for publication in TMLR.

Major issues raised by the reviewers were:
* Strong theoretical assumptions for the closed-form solution, which will not hold in practice. Authors discussed this and presented an empirical analysis of how much impact the violation of these assumptions has in practice.
* Nature of robustness and undesired embeddings and their relation to practical adversarial attacks not entirely clear. Discussed by authors in their responses.
* Questions around the difference to closely related previous work. Clarified by authors in their responses.
* Relevance to trustworthy ML and removing bias seem overstated. Addressed hypothetically by authors, but the usefulness of the method to remove model bias in practice is not shown in the paper (and I personally would consider that beyond the scope of this work).
* Concerns about size of "large" language models (orig. publication had models with hundreds of millions of parameters) and the models being somewhat outdated. Authors added larger models (~1B parameters) during rebuttal. The reviewer that raised this concerns thinks that the concern still somewhat stands and suggests removing the term "large language models" from the title. I personally agree somewhat, but have no strong reason to believe that the method would behave qualitatively very differently at larger model scale. The term "large" will likely not stand the test of time either way, and as long as the paper (early on) clarifies the scale of models used, I personally think this is fine.

**Audience:**

All reviewers and the AE agree that part of TMLR's audience is interested in the findings of this paper.

**Claims And Evidence:**

After the authors' responses and based on the improved manuscript, all reviewers argue that the claims made in the submission are supported by accurate, convincing, and clear evidence, with which I agree.

To the authors: note that there is still some mild criticism regarding the broadness of the claims, the relevance wrt. a range of practical adversarial attacks, and one reviewer argues that the use of "large" language models in the title is somewhat of an overstatement. I personally do not think we need another round of revisions / comments, and all reviewers lean towards acceptance; I therefore post these comments from the reviewer recommendation for the authors' consideration to perhaps do a final pass over the introduction and wording of the main claims.